# GENERALIZED CONSISTENCY TRAJECTORY MODELS FOR IMAGE MANIPULATION

**Beomsu Kim**[*]**, Jaemin Kim**[*]**, Jeongsol Kim & Jong Chul Ye**
KAIST
{beomsu.kim, kjm981995, jeongsol, jong.ye}@kaist.ac.kr

## ABSTRACT

Diffusion models (DMs) excel in unconditional generation, as well as on applications such as image editing and restoration. The success of DMs lies in the iterative nature of diffusion: diffusion breaks down the complex process of mapping noise to data into a sequence of simple denoising tasks. Moreover, we are able to exert fine-grained control over the generation process by injecting guidance terms into each denoising step. However, the iterative process is also computationally intensive, often taking from tens up to thousands of function evaluations. Although consistency trajectory models (CTMs) enable traversal between any time points along the probability flow ODE (PFODE) and score inference with a single function evaluation, CTMs only allow translation from Gaussian noise to data. This work aims to unlock the full potential of CTMs by proposing generalized CTMs (GCTMs), which translate between arbitrary distributions via ODEs. We discuss the design space of GCTMs and demonstrate their efficacy in various image manipulation tasks such as image-to-image translation, restoration, and editing. Code is available at https://github.com/1202kbs/GCTM.

## 1 INTRODUCTION

Diffusion-based generative models (DMs) learn the scores of noise-perturbed data distributions, which can be used to translate samples between two distributions by numerically integrating an SDE or a probability flow ODE (PFODE) (Ho et al., 2020; Dhariwal & Nichol, 2021; Song et al., 2021). They have achieved remarkable progress over recent years, even surpassing well-known generative models such as Generative Adversarial Networks (GANs) (Goodfellow et al., 2014) or Variational Autoencoders (VAEs) (Kingma & Welling, 2014) in terms of sample quality. Moreover, diffusion models have found wide application in areas such as image-to-image translation (Saharia et al., 2022), image restoration (Chung et al., 2022; 2023), image editing (Meng et al., 2022), etc.

The success of DMs can largely be attributed to the iterative nature of diffusion, arising from its foundation on differential equations – multi-step generation grants high-quality image synthesis by breaking down the complex process of mapping noise to data into a composition of simple denoising steps. We are also able to exert fine-grained control over the generation process by injecting minute guidance terms into each step (Chung et al., 2022; Ho & Salimans, 2022). Indeed, guidance is an underlying principle behind numerous diffusion-based image editing and restoration algorithms.

However, its iterative nature is also a curse, as diffusion inference often demands from tens to thousands of number of neural function evaluations (NFEs) per sample, rendering practical usage difficult. Consequently, there is now a large body of works on improving the inference speed of DMs. Among them, distillation refers to methods which train a neural network to translate samples along PFODE trajectories generated by a pre-trained teacher DM in one or two NFEs. Representative distillation methods include progressive distillation (PD) (Salimans & Ho, 2022), consistency models (CMs) (Song et al., 2023), and consistency trajectory models (CTMs) (Kim et al., 2024b).

In contrast to PD or CMs which only allow traversal to the terminal point of the PFODE, CTMs enable traversal between any pair of time points along the PFODE as well as score inference, all in a single inference step. Thus, in theory, CTMs are more amenable to guidance, and are applicable to a wider variety of downstream image manipulation tasks. Yet, there is a lack of works exploring the effectiveness of CTMs in such context.

---

[*]Equal Contribution

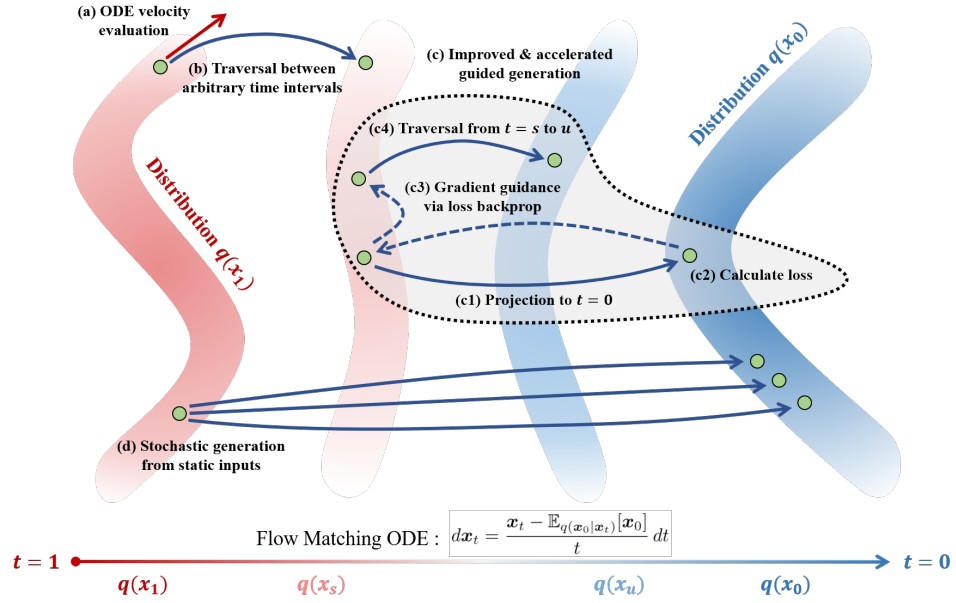

Figure 1: **An illustration of GCTM and its applications – solid arrows can be implemented by a single forward pass of the GCTM network.** GCTMs learn to traverse the Flow Matching ODE which is capable of *interpolating two arbitrary distributions* $q(\boldsymbol{x}_0)$ *and* $q(\boldsymbol{x}_1)$. GCTMs allow **(a)** one-step inference of ODE velocity, **(b)** one-step traversal between arbitrary time intervals of the ODE, **(c)** improved gradient-guidance by using exact posterior sample instead of posterior mean, and **(d)** one-step generation of varying outputs $\boldsymbol{x}_0$ given a fixed input $\boldsymbol{x}_1$.

In this work, we take a step towards unlocking the full potential of CTMs. To this end, we first propose generalized CTMs (GCTMs) which generalize the theoretical framework behind CTMs with Flow Matching (Lipman et al., 2023; Pooladian et al., 2023; Tong et al., 2024) to enable translation between *two arbitrary distributions*. Next, we discuss the design space of GCTMs, and how each design choice influences the downstream task performance. Finally, we demonstrate the power of GCTMs on a variety of image manipulation tasks. Our contributions can be summarized as follows.

- **Generalization of theory.** We propose GCTMs, which uses conditional flow matching theory to enable one-step translation between two arbitrary distributions (Theorem 1). This stands in contrast to CTMs, which is only able to learn PFODEs from Gaussian to data. In fact, we prove CTM is a special case of GCTM when one side is Gaussian (Theorem 2).
- **Elucidation of design space.** We clarify the design components of GCTMs, and explain how each component affects downstream task performance (Section 4.1). In particular, flexible choice of couplings enable GCTM training in both unsupervised and supervised settings, allowing us to accelerate zero-shot and supervised image manipulation algorithms.
- **Empirical verification.** We demonstrate the potential of GCTMs on unconditional generation, image-to-image translation, image restoration, image editing, and latent manipulation. We show that GCTMs achieve competitive performance even with NFE = 1.

## 2 RELATED WORK

**Diffusion model distillation.** Despite the success of diffusion models (DMs) in generation tasks, DMs require large number of function evaluations (NFEs). As a way to improve the inference speed, the distillation method is proposed to predict the previously trained teacher DM's output, *e.g.*, score function. Progressive distillation (PD) (Salimans & Ho, 2022) progressively reduces the NFEs by training the student model to learn predictions corresponding to two-steps of the teacher model's deterministic sampling path. Consistency models (CMs) (Song et al., 2023) perform distillation by reducing the self-consistency function over the generative ODE. The above methodologies only consider the output of the ODE path. In contrast, consistency trajectory model (CTM) (Kim et al., 2024b) simultaneously learns the integral and infinitesimal changes of the PFODE trajectory. Our paper extends CTM to learn the PFODE trajectory between two arbitrary distributions.

**Zero-shot image restoration via diffusion.** Image restoration such as super-resolution, deblurring, and inpainting can be formulated as inverse problems, which obtain true signals from given observations. With the advancements in DMs serving as powerful priors, diffusion based inverse solvers have been explored actively. DDRM (Kawar et al., 2022) performs denoising steps on the spectral space of a linear corrupting matrix. DPS (Chung et al., 2022) and ΠGDM (Song et al., 2022) propose posterior sampling by estimating the likelihood distribution through Jensen's approximation and Gaussian assumption, respectively. While diffusion-based inverse solvers facilitate zero-shot image restoration, they often need prolonged sampling times. CoSIGN (Zhao et al., 2024) addresses this problem by using CMs as generative priors, but we show that GCTMs can be better priors.

**Image translation via diffusion.** The seminal work Pix2Pix (Isola et al., 2017) achieves image-to-image translation with conditional GANs. SDEdit (Meng et al., 2022) avoids mode collapse and learning instabilities with GANs by utilizing DMs to translate edited images along SDEs. Palette (Saharia et al., 2022) proposed conditional DMs for image-to-image translation tasks. To address the Gaussian prior constraint with DMs, Schrödinger bridge (SB) (Liu et al., 2023; Kim et al., 2024a), direct diffusion bridge (DDB) (Delbracio & Milanfar, 2023), or denoising diffusion implicit bridge (DDIB) (Su et al., 2023) methods have been proposed to learn SDEs or ODEs between arbitrary two distributions. However, such models often require large NFEs. This has inspired models which distill conditional ODE trajectories (Mei et al., 2024; Xiao et al., 2024), DDB trajectories (He et al., 2024), or DDIB trajectories (Starodubcev et al., 2024). GCTMs are more general in the sense that they enable velocity evaluation and translation between two arbitrary timesteps.

## 3 BACKGROUND

### 3.1 DIFFUSION MODELS (DMs)

DMs (Song et al., 2021; Ho et al., 2020) learn to reverse the process of corrupting data into Gaussian noise. Formally, the corruption process can be described by a forward SDE

$$d\boldsymbol{x}_\tau = \sqrt{2\tau}\, d\boldsymbol{w}_\tau \tag{1}$$

defined on the time interval $\tau \in (0, \infty)$. Given $\boldsymbol{x}_0$ distributed according to a data distribution $p(\boldsymbol{x}_0)$, (1) sends $\boldsymbol{x}_0$ to Gaussian noise as $\tau$ increases from 0 to $\infty$. The reverse of the corruption process can be described by the reverse SDE

$$d\boldsymbol{x}_\tau = -2\tau \nabla \log p(\boldsymbol{x}_\tau)\, d\tau + \sqrt{2\tau}\, d\bar{\boldsymbol{w}}_\tau \tag{2}$$

or its deterministic counterpart, the probability flow ODE (PFODE)

$$d\boldsymbol{x}_\tau = -\tau \nabla \log p(\boldsymbol{x}_\tau)\, d\tau = \tau^{-1}(\boldsymbol{x}_\tau - \mathbb{E}_{p(\boldsymbol{x}_0|\boldsymbol{x}_\tau)}[\boldsymbol{x}_0])\, d\tau \tag{3}$$

where $p(\boldsymbol{x}_\tau)$ is the distribution of $\boldsymbol{x}_\tau$ following (1), and $\bar{\boldsymbol{w}}_\sigma$ is the standard Wiener process in reverse-time. Given noise $\boldsymbol{x}_{\hat{\tau}} \sim \mathcal{N}(\boldsymbol{x}_{\hat{\tau}}|\boldsymbol{0}, \hat{\tau}^2 \boldsymbol{I}) \approx p(\boldsymbol{x}_{\hat{\tau}})$ for some large $\hat{\tau}$, $\boldsymbol{x}_\tau$ following (2) or (3) is distributed $p(\boldsymbol{x}_\tau)$ as $\tau$ decreases from $\hat{\tau}$ to 0. Thus, DMs are able to generate data from noise by approximating the scores $\nabla \log p(\boldsymbol{x}_\tau)$ via score matching and numerically integrating (2) or (3).

### 3.2 CONSISTENCY TRAJECTORY MODELS (CTMs)

CTMs (Kim et al., 2024b) learn to translate samples between arbitrary time points of PFODE trajectories, *i.e.*, the goal of CTMs is to learn the integral of the PFODE

$$G(\boldsymbol{x}_\tau, \tau, \sigma) \coloneqq \boldsymbol{x}_\tau + \int_\tau^\sigma u^{-1}(\boldsymbol{x}_u - \mathbb{E}_{p(\boldsymbol{x}_0|\boldsymbol{x}_u)}[\boldsymbol{x}_0])\, du \tag{4}$$

for $\sigma \leq \tau$, where the terminal distribution $p(\boldsymbol{x}_{\hat{\sigma}})$ is assumed to be Gaussian. The parametrization

$$\begin{cases} G(\boldsymbol{x}_\tau, \tau, \sigma) = \frac{\sigma}{\tau}\boldsymbol{x}_\tau + \left(1 - \frac{\sigma}{\tau}\right) g(\boldsymbol{x}_\tau, \tau, \sigma) \\ g(\boldsymbol{x}_\tau, \tau, \sigma) \coloneqq \boldsymbol{x}_\tau + \frac{\tau}{\tau - \sigma} \int_\tau^\sigma u^{-1}(\boldsymbol{x}_u - \mathbb{E}_{p(\boldsymbol{x}_0|\boldsymbol{x}_u)}[\boldsymbol{x}_0])\, du \end{cases} \tag{5}$$

enables both traversal along the PFODE as well as score inference, since

$$\lim_{\sigma \to \tau} g(\boldsymbol{x}_\tau, \tau, \sigma) = \mathbb{E}_{p(\boldsymbol{x}_0|\boldsymbol{x}_\tau)}[\boldsymbol{x}_0] \tag{6}$$

so we may define $g(\boldsymbol{x}_\tau, \tau, \tau) \coloneqq \mathbb{E}_{p(\boldsymbol{x}_0|\boldsymbol{x}_\tau)}[\boldsymbol{x}_0]$.

Given a pre-trained DM, CTMs approximate $g$ with a neural net $g_\theta$ by simultaneously minimizing a distillation loss and a denoising score matching (DSM) loss. The distillation loss is

$$\mathcal{L}_{\text{CTM}}(\theta) \coloneqq \mathbb{E}_{0 \leq \sigma \leq u < \tau \leq \hat{\sigma}} \mathbb{E}_{p(\boldsymbol{x}_\tau)} \left[ d \left( G_\theta(\boldsymbol{x}_\tau, \tau, \sigma), G_{\text{sg}(\theta)}(\boldsymbol{x}_{\tau \to u}, u, \sigma) \right) \right] \tag{7}$$

where $G_\theta$ is the $G$-function with $g_\theta$ in place of $g$, $d(\cdot, \cdot)$ is a measure of similarity between inputs, sg is the stop-gradient operation, and $\boldsymbol{x}_{\tau \to u}$ is the integral of the PFODE from time $\tau$ to $u$ starting from $\boldsymbol{x}_\tau$ using score estimates from the pre-trained diffusion model. Minimization of (7) causes $G_\theta$ to adhere to PFODE trajectories generated by the pre-trained diffusion model. The DSM loss is

$$\mathcal{L}_{\text{DSM}}(\theta) \coloneqq \mathbb{E}_{0 \leq \tau \leq \hat{\tau}} \mathbb{E}_{p(\boldsymbol{x}_0) \mathcal{N}(\boldsymbol{\epsilon}|\boldsymbol{0}, \boldsymbol{I})} \mathbb{E}_{p(\boldsymbol{x}_\tau|\boldsymbol{x}_0, \boldsymbol{\epsilon})} \left[ \|\boldsymbol{x}_0 - g_\theta(\boldsymbol{x}_\tau, \tau, \tau)\|_2^2 \right] \tag{8}$$

where $p(\boldsymbol{x}_\tau|\boldsymbol{x}_0, \boldsymbol{\epsilon}) = \delta_{\boldsymbol{x}_0 + \tau \boldsymbol{\epsilon}}(\boldsymbol{x}_\tau)$ and $\delta_{\boldsymbol{y}}(\cdot)$ is a Dirac delta at $\boldsymbol{y}$. Minimization of (8) causes $g_\theta$ to satisfy (6) (Vincent, 2011). This loss acts as a regularization which improves score accuracy, and is crucial for sampling with large NFEs (Kim et al., 2024b). Thus, the final CTM training objective is

$$\mathcal{L}_{\text{CTM}}(\theta) + \lambda_{\text{DSM}} \mathcal{L}_{\text{DSM}}(\theta), \tag{9}$$

and it is possible to further improve sample quality by adding a GAN loss.

## 3.3 Flow Matching (FM)

FM (Lipman et al., 2023; Tong et al., 2024; Pooladian et al., 2023) is another technique for learning PFODEs between two distributions $q(\boldsymbol{x}_0)$ and $q(\boldsymbol{x}_1)$. Specifically, let $q(\boldsymbol{x}_0, \boldsymbol{x}_1)$ be a joint distribution of $q(\boldsymbol{x}_0)$ and $q(\boldsymbol{x}_1)$. Define

$$q(\boldsymbol{x}_t|\boldsymbol{x}_0, \boldsymbol{x}_1) \coloneqq \delta_{(1-t)\boldsymbol{x}_0 + t\boldsymbol{x}_1}(\boldsymbol{x}_t), \quad q(\boldsymbol{x}_t) \coloneqq \mathbb{E}_{q(\boldsymbol{x}_0, \boldsymbol{x}_1)}[q(\boldsymbol{x}_t|\boldsymbol{x}_0, \boldsymbol{x}_1)] \tag{10}$$

where $t \in (0, 1)$. Then, by Theorem 3.1 in Tong et al. (2024), the ODE given by

$$d\boldsymbol{x}_t = \mathbb{E}_{q(\boldsymbol{x}_0, \boldsymbol{x}_1|\boldsymbol{x}_t)}[\boldsymbol{x}_1 - \boldsymbol{x}_0] \, dt \tag{11}$$

generates the probability path $q(\boldsymbol{x}_t)$, i.e., with terminal condition $\boldsymbol{x}_1 \sim q(\boldsymbol{x}_1)$, $\boldsymbol{x}_t$ following (11) is distributed according to $q(\boldsymbol{x}_t)$. Analogous to DSM, the velocity term in (11) can be approximated by a neural network $\boldsymbol{v}_\phi$ which solves a regression problem (see Theorem 3.2 in Tong et al. (2024))

$$\min_\phi \mathbb{E}_{q(\boldsymbol{x}_0, \boldsymbol{x}_1, \boldsymbol{x}_t)} \left[ \|(\boldsymbol{x}_1 - \boldsymbol{x}_0) - \boldsymbol{v}_\phi(\boldsymbol{x}_t, t)\|_2^2 \right]. \tag{12}$$

However, unlike diffusion whose terminal distribution $p(\boldsymbol{x}_{\hat{\sigma}})$ is Gaussian, $q(\boldsymbol{x}_1)$ can be arbitrary. We provide a complete proof of correctness of this section in Appendix C.1.

## 4 Generalized Consistency Trajectory Models (GCTMs)

We now present GCTMs, which generalize CTMs to enable translation between arbitrary distributions. We begin with a crucial proposition which proves we can parametrize the solution to the FM ODE (11) in a form analogous to CTMs. The proof is deferred to Appendix C.2.

**Proposition 1.** *The ODE (11) is equivalent to*

$$d\boldsymbol{x}_t = t^{-1}(\boldsymbol{x}_t - \mathbb{E}_{q(\boldsymbol{x}_0|\boldsymbol{x}_t)}[\boldsymbol{x}_0]) \, dt \tag{13}$$

*defined on $t \in (0, 1)$. Hence, we can express the solution to (11) as*

$$\begin{cases} G(\boldsymbol{x}_t, t, s) = \frac{s}{t} \boldsymbol{x}_t + \left(1 - \frac{s}{t}\right) g(\boldsymbol{x}_t, t, s), \\ g(\boldsymbol{x}_t, t, s) \coloneqq \boldsymbol{x}_t + \frac{t}{t-s} \int_t^s u^{-1}(\boldsymbol{x}_u - \mathbb{E}_{q(\boldsymbol{x}_0|\boldsymbol{x}_u)}[\boldsymbol{x}_0]) \, du. \end{cases} \tag{14}$$

There are two differences between (5) and (14). First, the time variables $t$ and $s$ now lie in the unit interval $(0, 1)$ instead of $(0, \infty)$, and second, $p(\boldsymbol{x}_0|\boldsymbol{x}_u)$ is replaced with $q(\boldsymbol{x}_0|\boldsymbol{x}_u)$. The second difference is what enables translation between arbitrary distributions, as $q(\boldsymbol{x}_0|\boldsymbol{x}_u)$ recovers clean images $\boldsymbol{x}_0$ given images $\boldsymbol{x}_u$ perturbed by arbitrary type of vectors (e.g., Gaussian noise, images, etc.), while $p(\boldsymbol{x}_0|\boldsymbol{x}_u)$ recovers clean images $\boldsymbol{x}_0$ only for Gaussian-perturbed samples $\boldsymbol{x}_u$. We call a neural network $g_\theta$ which approximates $g$ in (14) a GCTM, and we can train such a network by optimizing the FM counterparts of $\mathcal{L}_{\text{CTM}}$ and $\mathcal{L}_{\text{DSM}}$:

$$\mathcal{L}_{\text{GCTM}}(\theta) \coloneqq \mathbb{E}_{0 \leq s \leq u < t \leq 1} \mathbb{E}_{q(\boldsymbol{x}_t)} \left[ d \left( G_\theta(\boldsymbol{x}_t, t, s), G_{\text{sg}(\theta)}(\boldsymbol{x}_{t \to u}, u, s) \right) \right], \tag{15}$$

$$\mathcal{L}_{\text{FM}}(\theta) \coloneqq \mathbb{E}_{0 \leq t \leq 1} \mathbb{E}_{q(\boldsymbol{x}_0, \boldsymbol{x}_1)} \mathbb{E}_{q(\boldsymbol{x}_t|\boldsymbol{x}_0, \boldsymbol{x}_1)} \left[ \|\boldsymbol{x}_0 - g_\theta(\boldsymbol{x}_t, t, t)\|_2^2 \right]. \tag{16}$$

The next proposition shows that the PFODE (3) learned by CTMs is a special case of the ODE (13) learned by GCTMs, so GCTMs indeed generalize CTMs. The proof is deferred to Appendix C.3.

**Proposition 2.** *Consider the choice of $q(\boldsymbol{x}_0, \boldsymbol{x}_1) = p(\boldsymbol{x}_0) \cdot \mathcal{N}(\boldsymbol{x}_1 | \boldsymbol{0}, \boldsymbol{I})$. Let*

$$t := \tau/(1+\tau), \qquad \boldsymbol{x}_t := \boldsymbol{x}_\tau/(1+\tau) \tag{17}$$

*where $\tau \in (0, \infty)$ and $\boldsymbol{x}_\tau$ follows the PFODE (3). Then*

$$\mathbb{E}_{p(\boldsymbol{x}_0 | \boldsymbol{x}_\tau)}[\boldsymbol{x}_0] = \mathbb{E}_{q(\boldsymbol{x}_0 | \boldsymbol{x}_t)}[\boldsymbol{x}_0] \tag{18}$$

*and $\boldsymbol{x}_t$ follows the ODE*

$$d\boldsymbol{x}_t = t^{-1}(\boldsymbol{x}_t - \mathbb{E}_{q(\boldsymbol{x}_0 | \boldsymbol{x}_t)}[\boldsymbol{x}_0]) \, dt \tag{19}$$

*on $t \in (0, 1)$. Furthermore, let $G_{\mathrm{CTM}}$, $g_{\mathrm{CTM}}$ denote CTM solutions and let $G_{\mathrm{GCTM}}$, $g_{\mathrm{GCTM}}$ denote GCTM solutions. Then with $s = \sigma/(1 + \sigma)$,*

$$\begin{cases} G_{\mathrm{CTM}}(\boldsymbol{x}_\tau, \tau, \sigma) = G_{\mathrm{GCTM}}(\boldsymbol{x}_t, t, s) \cdot (1 + s), \\ g_{\mathrm{CTM}}(\boldsymbol{x}_\tau, \tau, \tau) = g_{\mathrm{GCTM}}(\boldsymbol{x}_t, t, t). \end{cases} \tag{20}$$

In short, (18) shows the equivalence of scores, and (19) shows the equivalence of ODEs. Thus, given $g_\theta$ trained with $\mathcal{L}_{\mathrm{FM}}$ and $\mathcal{L}_{\mathrm{GCTM}}$ with the setting of Prop. 2, we are able to evaluate diffusion scores and simulate diffusion PFODE trajectories with a simple change of variables (17), as shown in (20).

Given GCTM's capability to replicate CTM, we will now outline the key components of GCTM that enable its significant extension for various downstream tasks. This flexibility offers a notable advantage of GCTM over CTM.

## 4.1 THE DESIGN SPACE OF GCTMS

**Coupling** $q(\boldsymbol{x}_0, \boldsymbol{x}_1)$. In contrast to diffusion which only uses the trivial coupling $q(\boldsymbol{x}_0, \boldsymbol{x}_1) = q(\boldsymbol{x}_0)q(\boldsymbol{x}_1)$ in $\mathcal{L}_{\mathrm{DSM}}(\theta)$, FM allows us to use arbitrary joint distributions of $q(\boldsymbol{x}_0)$ and $q(\boldsymbol{x}_1)$ in $\mathcal{L}_{\mathrm{FM}}(\theta)$. Intuitively, $q(\boldsymbol{x}_0, \boldsymbol{x}_1)$ encodes our inductive bias for what kind of pairs $(\boldsymbol{x}_0, \boldsymbol{x}_1)$ we wish the model to learn, since FM ODE is distributed $q(\boldsymbol{x}_t)$ at each time $t$, and $q(\boldsymbol{x}_t)$ is the distribution of $(1-t)\boldsymbol{x}_0 + t\boldsymbol{x}_1$ for $(\boldsymbol{x}_0, \boldsymbol{x}_1) \sim q(\boldsymbol{x}_0, \boldsymbol{x}_1)$. Here, we list three valid couplings of GCTM as examples (see Alg. 1 for code). In contrast, CTM only uses a special case of the independent coupling.

- *Independent coupling*:

$$q(\boldsymbol{x}_0, \boldsymbol{x}_1) = q(\boldsymbol{x}_0)q(\boldsymbol{x}_1) \tag{21}$$

  This coupling reflects no prior assumption about the relation between $\boldsymbol{x}_0$ and $\boldsymbol{x}_1$. As shown earlier, diffusion models use this type of coupling with standard normal $q(\boldsymbol{x}_1)$.

- *Minibatch entropic optimal transport (EOT) coupling*: in practice, FM loss (16) is approximated by an average over minibatch of pairs $(\boldsymbol{x}_0, \boldsymbol{x}_1)$. We can consider minibatch EOT coupling samples (Pooladian et al., 2023) which are generated by sampling $\{\boldsymbol{x}_0^i\}_{i=1}^K$ from $q(\boldsymbol{x}_0)$, sampling $\{\boldsymbol{x}_1^i\}_{i=1}^K$ from $q(\boldsymbol{x}_1)$, running the Sinkhorn algorithm (Cuturi, 2013) (see Alg. 3) to create a doubly-stochastic EOT matrix $\boldsymbol{P}^{\mathrm{EOT}}$ between the two batches, and sampling $(\boldsymbol{x}_0, \boldsymbol{x}_1)$ pairs from $\boldsymbol{P}^{\mathrm{EOT}}$. As observed by Pooladian et al. (2023), minibatch EOT coupling can accelerate flow matching optimization by reducing gradient variance, and we expect similar benefits for GCTM training as well.

- *Supervised coupling*:

$$q(\boldsymbol{x}_0, \boldsymbol{x}_1) = \int q(\boldsymbol{x}_0)q(\boldsymbol{H}|\boldsymbol{x}_0)\delta_{\boldsymbol{H}\boldsymbol{x}_0}(\boldsymbol{x}_1) \, d\boldsymbol{H} \tag{22}$$

  where $\boldsymbol{H} \sim q(\boldsymbol{H}|\boldsymbol{x}_0)$ is a random operator, possibly dependent on $\boldsymbol{x}_0$, which maps ground-truth data $\boldsymbol{x}_0$ to observations $\boldsymbol{x}_1$, *i.e.*, $\boldsymbol{x}_1 = \boldsymbol{H}\boldsymbol{x}_0$. For instance, in the context of learning an inpainting model, $\boldsymbol{H}$ is could be a random masking operator.

**Gaussian perturbation.** The cardinality of the support of $q(\boldsymbol{x}_1)$ must be larger than or equal to the cardinality of the support of $q(\boldsymbol{x}_0)$ for there to be a well-defined ODE from $q(\boldsymbol{x}_1)$ to $q(\boldsymbol{x}_0)$. This is because the ODE trajectory given an initial condition is unique, so a single sample $\boldsymbol{x}_1 \sim q(\boldsymbol{x}_1)$ cannot be transported to multiple points in the support of $q(\boldsymbol{x}_0)$. A simple way to address this problem is to add small Gaussian noise to $q(\boldsymbol{x}_1)$ samples such that $q(\boldsymbol{x}_1)$ is supported everywhere.

---

**Algorithm 1** $q(\boldsymbol{x}_0, \boldsymbol{x}_1)$ Sampling

1: **Assume** $m = 1, \ldots, M$, Batch size $M$
2: **if** Coupling is `Independent` **then**
3:      $\{\boldsymbol{x}_0^m\}_m \sim q(\boldsymbol{x}_0), \{\boldsymbol{x}_1^m\}_m \sim q(\boldsymbol{x}_1)$
4:      **Return** $\{(\boldsymbol{x}_0^m, \boldsymbol{x}_1^m)\}_m$
5: **else if** Coupling is `Minibatch EOT` **then**
6:      $\{\boldsymbol{x}_0^m\}_m \sim q(\boldsymbol{x}_0), \{\boldsymbol{x}_1^m\}_m \sim q(\boldsymbol{x}_1)$
7:      **Return** SK$(\{\boldsymbol{x}_0^m\}_m, \{\boldsymbol{x}_1^m\}_m, \tau)$
8: **else if** Coupling is `Supervised` **then**
9:      $\{\boldsymbol{x}_0^m\}_m \sim q(\boldsymbol{x}_0), \boldsymbol{H}^m \sim q(\boldsymbol{H}|\boldsymbol{x}_0^m)$
10:      **Return** $\{(\boldsymbol{x}_0^m, \boldsymbol{H}^m \boldsymbol{x}_0^m)\}_m$
11: **end if**

---

**Algorithm 2** GCTM Training

1: **while** training **do**
2:      Sample times $\{\hat{t}^m\}_m, \{(t^m, s^m, u^m)\}_m$
3:      With Alg. 1, $\{(\boldsymbol{x}_0^m, \boldsymbol{x}_1^m)\}_m \sim q(\boldsymbol{x}_0, \boldsymbol{x}_1)$
4:      $\boldsymbol{x}_{\hat{t}^m}^m \leftarrow (1 - \hat{t}^m)\boldsymbol{x}_0^m + \hat{t}^m \boldsymbol{x}_1^m$
5:      $\boldsymbol{x}_{t^m}^m \leftarrow (1 - t^m)\boldsymbol{x}_0^m + t^m \boldsymbol{x}_1^m$
6:      $\mathcal{L}_{\mathrm{FM}}(\theta) = \frac{1}{M}\sum_m \|\boldsymbol{x}_0^m - g_\theta(\boldsymbol{x}_{\hat{t}^m}^m, \hat{t}^m, \hat{t}^m)\|_2^2$
7:      $\widetilde{\boldsymbol{x}}_{s^m}^m \leftarrow G_{\mathrm{sg}(\theta)}(\boldsymbol{x}_{t^m \rightarrow u^m}^m, u^m, s^m)$
8:      $\mathcal{L}_{\mathrm{GCTM}}(\theta) = \frac{1}{M}\sum_{m=1}^M d(G_\theta(\boldsymbol{x}_{t^m}^m, t^m, s^m), \widetilde{\boldsymbol{x}}_{s^m}^m)$
9:      Minimize $\mathcal{L}_{\mathrm{GCTM}}(\theta) + \lambda_{\mathrm{FM}}\mathcal{L}_{\mathrm{FM}}(\theta)$
10: **end while**

---

We emphasize that Gaussian perturbation allows GCTMs to achieve one-to-many generation when we use the supervised coupling. Concretely, consider the scenario where there are multiple labels $\boldsymbol{x}_0 \sim q(\boldsymbol{x}_0|\boldsymbol{x}_1)$ which correspond to an observed $\boldsymbol{x}_1$. Then, the perturbation $\epsilon$ added to $\boldsymbol{x}_1$ acts as a source of randomness, allowing the GCTM network to map $\boldsymbol{x}_1 + \epsilon$ to distinct labels $\boldsymbol{x}_0$ for distinct $\epsilon$. This stands in contrast to simply regressing the neural network output of $\boldsymbol{x}_1$ to corresponding labels $\boldsymbol{x}_0 \sim q(\boldsymbol{x}_0|\boldsymbol{x}_1)$ with $\ell_2$ loss, as this will cause the network to map $\boldsymbol{x}_1$ to the blurry posterior mean $\mathbb{E}_{q(\boldsymbol{x}_0|\boldsymbol{x}_1)}[\boldsymbol{x}_0]$ instead of a sharp image $\boldsymbol{x}_0$. Indeed, in Section 5.2, we observe blurry outputs if we use regression instead of GCTMs.

**Time discretization.** In practice, to optimize $\mathcal{L}_{\mathrm{GCTM}}(\theta)$, we sample time variables $s$, $t$, $u$ from a discretization $t_0 < t_1 < \cdots < t_N$ of the unit interval $[0, 1]$, and simulate $\boldsymbol{x}_{t \rightarrow u}$ with respect to the discretization as well. Given the success of the EDM time discretization Karras et al. (2022) for fast sampling of diffusion models, we propose using the EDM time discretization converted to FM time discretization via change of variables in Proposition 2,

$$t_0 = 0, \quad t_i = \frac{\tau_i}{\tau_i + 1} \quad \textit{where} \quad \tau_i = (\sigma_{\min}^{1/\rho} + \frac{i}{N}(\sigma_{\max}^{1/\rho} - \sigma_{\min}^{1/\rho}))^\rho. \tag{23}$$

We fix $\rho = 7$ and $\sigma_{\min} = 0.002$ as proposed in Karras et al. (2022), and control $\sigma_{\max}$. We note that $\sigma_{\max}$ controls the amount of emphasis on time near $t = 1$, *i.e.*, larger $\sigma_{\max}$ places more time discretization points near $t = 1$.

## 5 EXPERIMENTS

We now explore the possibilities of GCTMs on unconditional generation, image-to-image translation, image restoration, image editing, and latent manipulation. In particular, GCTM admits NFE = 1 sampling via $\boldsymbol{x}_t \mapsto G_\theta(\boldsymbol{x}_t, t, 0)$. Due to the similarities between CTMs and GCTMs as detailed in Thm. 1, GCTMs can be trained using CTM training methods. In fact, we run Alg. 2 with the method in Section 5.2 of (Kim et al., 2024b) to train all GCTMs without pre-trained teacher models. A complete description of training settings are deferred to Appendix A.

### 5.1 FAST UNCONDITIONAL GENERATION

In the scenario where we do not have access to data pairs, we must resort to either the independent coupling or the OT coupling. Here, we show that the optimal transport coupling can significantly accelerate the convergence speed of GCTMs during training, especially when we use a smaller number of timesteps $N$ in time discretization during GCTM training (see Section 4.1). Using small $N$ may be of interest when we wish to trade-off training speed for performance, since per-iteration training cost of GCTMs increases linearly with $N$. For instance, when $t = 1$ and $u = s = 0$ in the GCTM loss (15), we need to integrate along the entire time interval $(0, 1)$, which requires $N$ steps of ODE integration.

In Figure 2, we observe up to $\times 2.5$ acceleration in terms of training iterations when we use OT coupling instead of independent coupling. Indeed, in Figure 3, OT coupling samples are visually sharper than independent coupling samples. We postulate this is because (1) OT coupling leads to straighter

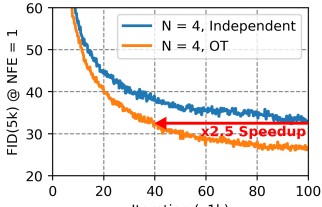

Figure 2: Training acceleration.

| Method | Teacher | FID $\downarrow$ |
|---|---|---|
| CTM | ✓ | 5.28 |
| | ✗ | 9.00 |
| CM | ✓ | 3.55 |
| | ✗ | 8.70 |
| iCM | ✗ | 2.51 |
| GCTM (OT) | ✗ | 5.32 |

Table 1: FID at NFE = 1.

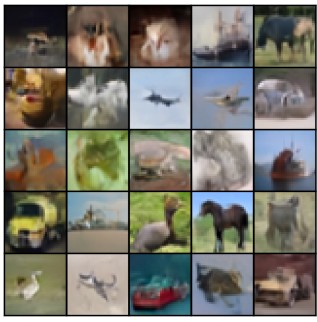 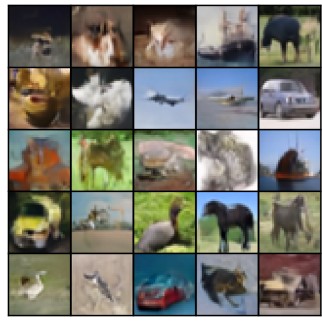 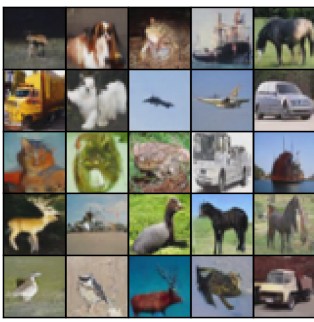

(a) Indep. $N = 4$, FID = 24.7          (b) OT $N = 4$, FID = 18.2          (c) OT $N = 32$, FID = 5.32

Figure 3: CIFAR10 unconditional samples with NFE = 1.

| Method | NFE | Time (ms) | Edges→Shoes | | | Night→Day | | | Facades | | |
|---|---|---|---|---|---|---|---|---|---|---|---|
| | | | FID ↓ | IS ↑ | LPIPS ↓ | FID ↓ | IS ↑ | LPIPS ↓ | FID ↓ | IS ↑ | LPIPS ↓ |
| Regression | 1 | 87 | 54.3 | 3.41 | 0.100 | 189.2 | 1.85 | 0.373 | 121.8 | **3.28** | 0.274 |
| Pix2Pix (Isola et al., 2017) | 1 | 33 | 77.0 | 3.17 | 0.208 | 158.0 | 1.68 | 0.418 | 134.1 | 2.74 | 0.288 |
| Palette (Saharia et al., 2022) | 5 | 166 | 334.1 | 1.90 | 0.861 | 350.2 | 1.16 | 0.707 | 259.3 | 2.47 | 0.394 |
| I²SB(Liu et al., 2023) | 5 | 284 | 53.9 | 3.23 | 0.154 | **145.8** | 1.79 | 0.376 | 135.2 | 2.51 | 0.269 |
| GCTM | 1 | 87 | **40.3** | **3.54** | **0.097** | 148.8 | **2.00** | **0.317** | **111.3** | 2.99 | **0.230** |

Table 2: I2I translation results ($64 \times 64$ resolution). Best is in **bold**, second best is underlined.

ODE trajectories, so we can accurately integrate ODEs with smaller $N$, and (2) lower variance from OT pairs leads to smaller variance in loss gradients, as discussed in (Pooladian et al., 2023).

In Table 1, we compare the Fréchet Inception Distance (FID) (Heusel et al., 2017) of GCTM and relevant baselines on CIFAR10 with NFE = 1. In the setting where we do not use a pre-trained teacher diffusion model, GCTM with OT coupling outperforms all methods with the exception of iCM (Song & Dhariwal, 2024), which is an improved variant of CM. Moreover, GCTM is on par with CTM trained with a teacher. Here, the numbers for CTM are our reproduced results without GAN loss for fair comparison. We speculate that further tuning of hyper-parameters or addition of a GAN loss could push the performance of GCTMs to match that of iCMs.

## 5.2 FAST IMAGE-TO-IMAGE TRANSLATION

Unlike previous distillation methods such as CM or CTM, GCTM can learn ODEs between arbitrary distributions, enabling image-to-image translation. To numerically validate this theoretical improvement, we train GCTMs on three translation tasks Edges→Shoes, Night→Day, and Facades (Isola et al., 2017), scaled to 64×64, with the supervised coupling. We consider three baseline methods: $\ell_2$-regression, Pix2Pix (Isola et al., 2017), Palette (Saharia et al., 2022) and I²SB(Liu et al., 2023). To evaluate translation performance, we use FID and Inception Score (IS) (Barratt & Sharma, 2018) to rate translation quality and LPIPS (Zhang et al., 2018) to assess faithfulness to input. We control NFEs such that all methods have similar inference times, and calculate all metrics on validation set.

In Table 2, we see GCTM shows strong performance on all tasks. SDE-based methods I²SB and Palette show poor performance at low NFEs, even when trained with pairs. Qualitative results in Figure 4 are in line with the metrics. Baselines produce blurry or nonsensical samples, while GCTM produces sharp and realistic images that are faithful to the input. In Table 5, we show results on higher resolution ($256 \times 256$) data, and observe similar trends.

## 5.3 FAST IMAGE RESTORATION

We consider two settings on the FFHQ $64 \times 64$ dataset, where we either know or do not know the corruption operator. In the former case, we train an unconditional GCTM with the independent coupling, with which we implement three zero-shot image restoration algorithms: DPS, CM-based image restoration, and the guided generation algorithm illustrated in Figure 1, where the loss is given as inconsistency between observations (see Append. B.2 for pseudo-codes and a detailed discussion of the differences). In the latter case, we train a GCTM with the supervised coupling

| Pix2Pix | Palette | I²SB | Regression | GCTM | $x_0$ | $x_1$ |

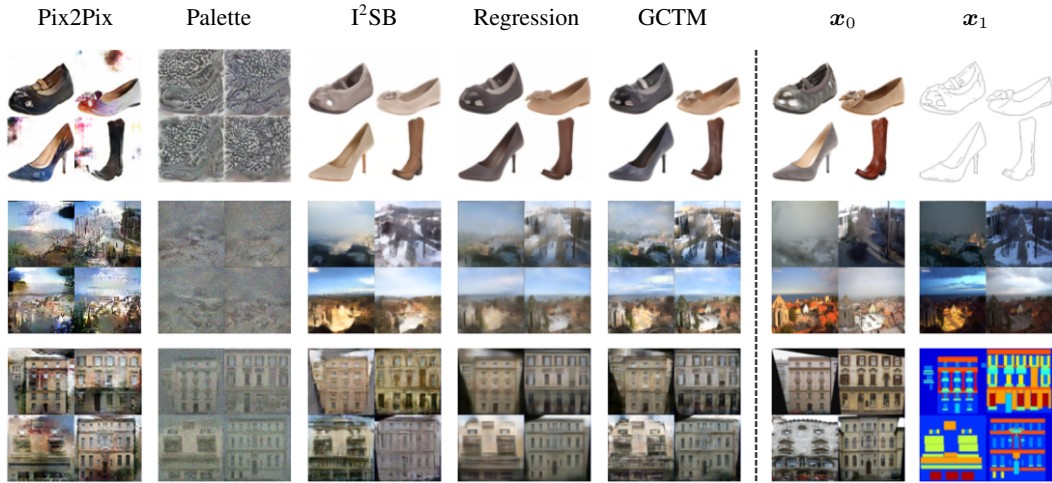

Figure 4: Qualitative evaluation of I2I translation $(64 \times 64$ resolution) on Edges→Shoes (top), Night→Day (middle) and Facades (bottom). NFE = 5 for I²SB and Palette.

| | Method | NFE | Time (ms) | SR2 - Bicubic | | | Deblur - Gaussian | | | Inpaint - Center | | |
|---|---|---|---|---|---|---|---|---|---|---|---|---|
| | | | | PSNR ↑ | SSIM ↑ | LPIPS ↓ | PSNR ↑ | SSIM ↑ | LPIPS ↓ | PSNR ↑ | SSIM ↑ | LPIPS ↓ |
| *0-Shot* | DPS | 32 | 1079 | 31.19 | 0.935 | 0.015 | 27.88 | 0.878 | 0.041 | **24.69** | **0.876** | **0.042** |
| | CM | 32 | 1074 | 30.80 | 0.930 | **0.010** | 27.85 | 0.871 | **0.027** | 23.02 | 0.857 | 0.050 |
| | GCTM | 32 | 1382 | **31.61** | **0.939** | 0.015 | **28.19** | **0.885** | 0.037 | 24.47 | **0.876** | **0.042** |
| *Superv.* | Regression | 1 | 87 | **33.46** | **0.964** | 0.015 | **31.19** | **0.942** | 0.015 | **28.76** | **0.922** | 0.028 |
| | Palette | 5 | 166 | 17.88 | 0.556 | 0.234 | 17.81 | 0.571 | 0.234 | 16.12 | 0.489 | 0.357 |
| | I²SB | 5 | 284 | 26.74 | 0.869 | 0.033 | 26.20 | 0.853 | 0.038 | 26.01 | 0.874 | 0.038 |
| | GCTM | 1 | 87 | 32.37 | 0.954 | **0.009** | 30.56 | 0.935 | **0.009** | 27.37 | 0.896 | **0.027** |

Table 3: Quantitative evaluation of image restoration on FFHQ $(64 \times 64$ resolution).

and $\ell_2$-regression, I²SB and Palette for comparison. Notably, GCTM is the only model applicable to both situations, thanks to the flexible choice of couplings. We again control NFEs such that all methods have similar inference speed.

Table 3 presents the numerical results in both settings. In the zero-shot setting, we see GCTM outperforming both DPS and CM. In particular, CM is slightly worse than DPS. Sample quality degradation due to error accumulation for CMs at large NFEs have already been observed in unconditional generation (*e.g.*, see Fig. 9 in (Kim et al., 2024b)), and we speculate a similar problem occurs for CMs in image restoration as well. On the other hand, GCTMs avoid this problem, as they are able to traverse to a smaller time using the ODE velocity approximated via $g_\theta$. In the supervised setting, we see regression attains the best PSNR and SSIM. This is a natural consequence of perception-distortion trade-off. Specifically, regression minimizes the MSE loss, so it leads to best distortion metrics (Delbracio & Milanfar, 2023) while producing blurry results. GCTM, which provides best results if we exclude regression on distortion

| $x_1$ | $x_0$ |

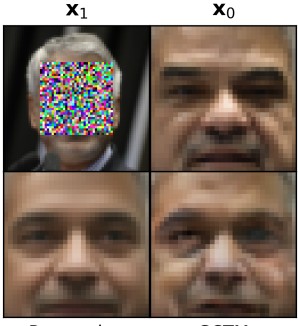

| Regression | GCTM |

Figure 5: Reg. vs. GCTM.

metrics (PSNR and SSIM) and best results on perception metrics (LPIPS), strikes the best balance between perception and distortion. For instance, in Fig. 5 inpainting results, regression sample lacks detail (e.g., wrinkles) while GCTM sample is sharp. We show more samples in Appendix E. In particular, in Table 6, we demonstrate image restoration task of GCTM on ImageNet with higher resolution $(256 \times 256$ resolution) images to demonstrate its scalability.

## 5.4 FAST IMAGE EDITING

In this section, we demonstrate that GCTM can perform realistic and faithful image editing without any special purpose training. Figure 6 shows image editing with an Edges→Shoes model and an unconditional FFHQ model. On Edges→Shoes, to edit an image, a user creates an edited input,

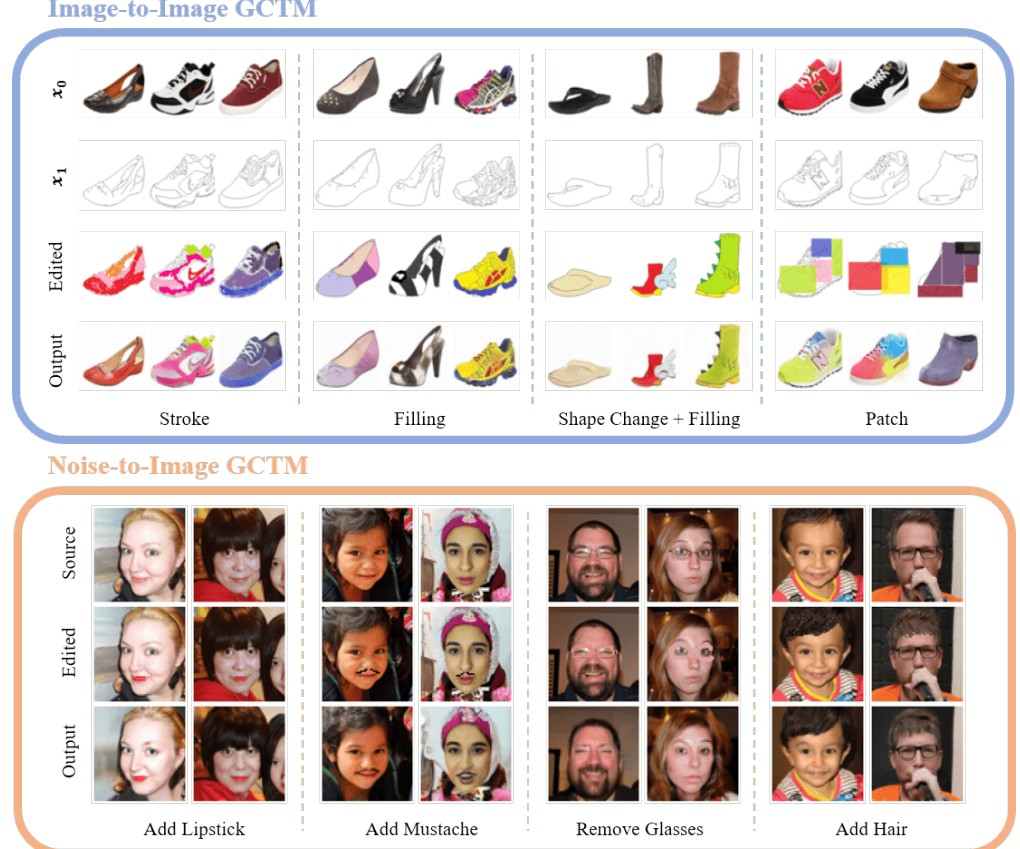

Figure 6: Image editing with GCTM, NFE = 1.

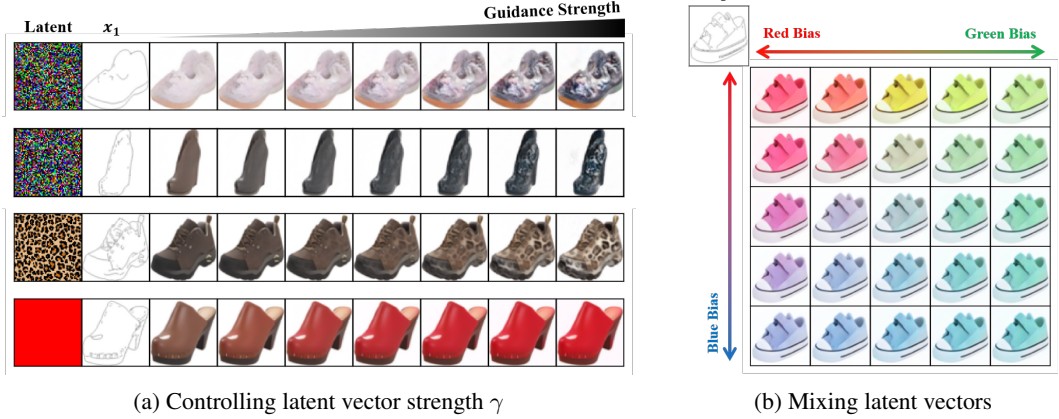

(a) Controlling latent vector strength $\gamma$      (b) Mixing latent vectors

Figure 7: Latent manipulation with image-to-image GCTM, NFE = 1.

which is an edge image painted to have a desired color and / or modified to have a desired outline. We then interpolate the edited input and the original edge image to a certain time point $t = s$ and send it to time $t = 0$ with GCTM to produce the output. On FFHQ, analogous to SDEdit (Meng et al., 2022), we interpolate an edited image with Gaussian noise and send it to time $t = 0$ with GCTM to generate the output. In contrast to previous image editing models such as SDEdit, GCTM requires only a single step to edit an image. Moreover, we observe that GCTM faithfully preserves source image structure while making the desired changes to the image.

## 5.5 FAST LATENT MANIPULATION

In this section, we demonstrate that GCTMs have a highly controllable latent space. Since there are plenty of works on latent manipulation with unconditional diffusion models, we focus on latent manipulation with GCTMs trained for image-to-image translation. For an image-to-image translation GCTM trained with Gaussian perturbation in Section 4.1, we assert that the perturbation added to $x_1$ can be manipulated to produce desired outputs $x_0$. In other words, the perturbation acts as a "latent vector" which controls the factors of variation in $x_0$. To test this hypothesis, in Figure 7, we display outputs $G_\theta(x_1 + \gamma\epsilon, 1, 0)$ for particular choices of $\epsilon$. In the left panel, we observe generated outputs increasingly adhere to the texture of latent $\epsilon$ as we increase guidance strength $\gamma$. Interestingly, GCTM generalizes well to latent vectors unseen during training, such as leopard spots or the color red. In the right panel, we explore the effect of linearly combining red, green, and blue latent vectors. We see that the desired color change is reflected faithfully in the outputs. These observations validate our hypothesis that image-to-image GCTMs have an interpretable latent space.

## 5.6 ABLATION STUDY

We now perform an ablation study on the design choices of Section 4.1. We have already illustrated the power of using appropriate couplings in previous sections, so we explore the importance of $\sigma_{\max}$. A robust choice for $\sigma_{\max}$ for unconditional generation is well-known to be $\sigma_{\max} = 80$ (Karras et al., 2022; Kim et al., 2024b), and we found using this choice to perform sufficiently well for GCTMs when learning to translate noise to data with independent or OT couplings. So, we restrict our attention to image-to-image translation.

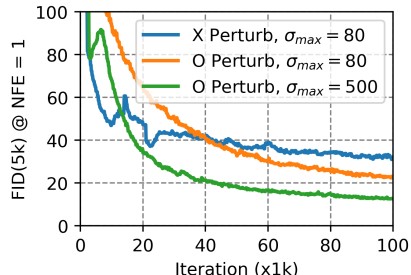

Figure 8: Ablation study of GCTM.

In Figure 8, we display the learning curves on Edges→Shoes for GCTMs trained without and with Gaussian perturbation, and $\sigma_{\max} \in \{80, 500\}$. We observe that GCTM trained without perturbation and $\sigma_{\max} = 80$ exhibits unstable dynamics, and is unable to minimize the FID below 30. On other hand, GCTM trained with perturbation and $\sigma_{\max} = 80$ surpasses the model trained without perturbation. This demonstrates Gaussian perturbation is indeed crucial for one-to-many generation, as noted in the last paragraph of Section 4.1. Finally, GCTM with both perturbation and $\sigma_{\max} = 500$ minimizes FID the fastest. This shows high-curvature regions for image-to-image ODEs lie near $x_1$, so we need to use a large $\sigma_{\max}$ which places more discretization points near $t = 1$.

## 6 CONCLUSION

Our work marks a significant advancement in the realm of ODE-based generative models, particularly on the transformative capabilities of Consistency Trajectory Models (CTMs). While the iterative nature of diffusion has proven to be a powerful foundation for high-quality image synthesis and nuanced control, the computational demands associated with numerous neural function evaluations (NFEs) per sample have posed challenges for practical implementation. Our proposal of Generalized CTMs (GCTMs) extends the reach of CTMs by enabling one-step translation between arbitrary distributions, surpassing the limitations of traditional CTMs confined to Gaussian noise to data transformations. Through an insightful exploration of the design space, we elucidate the impact of various components on downstream task performance, providing a comprehensive understanding that contributes to a broadly applicable and stable training scheme. Empirical validation across diverse image manipulation tasks demonstrates the potency of GCTMs, showcasing their ability to accelerate and enhance diffusion-based algorithms. In summary, our work not only contributes to theoretical advancements but also delivers tangible benefits, showcasing GCTMs as a key element in unlocking the full potential of diffusion models for practical, real-world applications in image synthesis, translation, restoration, and editing.

ACKNOWLEDGMENTS

This work was supported by the National Research Foundation of Korea under Grant RS-2024-00336454 and by the Institute for Information & Communications Technology Planning & Evaluation (IITP) grant funded by the Korea government (MSIT) (RS-2019-II190075, Artificial Intelligence Graduate School Program, KAIST).

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

# A  FULL EXPERIMENT SETTINGS

## A.1  TRAINING

In this section, we introduce training choices which provided reliable performance across all experiments in our paper.

**Bootstrapping scores.** In all our experiments, we train GCTMs without a pre-trained score model. So, analogous to CTMs, we use velocity estimates given by an exponential moving average $\theta_{\text{EMA}}$ of $\theta$ to solve ODEs. We use exponential moving average decay rate 0.999.

**Time discretization.** In practice, we discretize the unit interval into a finite number of timesteps $\{t_n\}_{n=0}^N$ where

$$t_0 = 0 < t_1 < \cdots < t_N = 1 \tag{24}$$

and learn ODE trajectories integrated with respect to the discretization schedule. EDM (Karras et al., 2022), which has shown robust performance on a variety of generation tasks, solves the PFODE on the time interval $(\sigma_{\min}, \sigma_{\max})$ for $0 < \sigma_{\min} < \sigma_{\max}$ according to the discretization schedule

$$\sigma_n = (\sigma_{\min}^{1/\rho} + (n/N)(\sigma_{\max}^{1/\rho} - \sigma_{\min}^{1/\rho}))^\rho \tag{25}$$

for $n = 0, \ldots, N$ and $\rho = 7$. Thus, using the change of time variable (17) derived in Theorem 1, we convert PFODE EDM schedule to FM ODE discretization

$$t_0 = 0, \quad t_n = \sigma_n/(1 + \sigma_n) \quad \text{for} \quad n = 1, \ldots, N-1, \quad t_N = 1. \tag{26}$$

In our experiments, we fix $\sigma_{\min} = 0.002$ and control $\sigma_{\max}$. We note that $\sigma_{\max}$ controls the amount of emphasis on time near $t = 1$, i.e., larger $\sigma_{\max}$ places more time discretization points near $t = 1$.

**Number of discretization steps $N$.** CTMs use fixed $N = 18$. In contrast, analogous to iCMs, we double $N$ every $100k$ iterations, starting from $N = 4$.

**Time $\hat{t}$ distribution.** For unconditional generation, we sample

$$t = \sigma/(1 + \sigma), \qquad \log \sigma \sim \mathcal{N}(-1.2, 1.2^2) \tag{27}$$

in accordance with EDM. For image-to-image translation, we sample

$$t \sim \text{beta}(3, 1). \tag{28}$$

**Network conditioning.** We use the EDM conditioning, following CTMs.

**Distance $d$.** CTMs use $d$ defined as

$$d(\boldsymbol{x}_t, \hat{\boldsymbol{x}}_t) = \text{LPIPS}(G_{\theta_{\text{EMA}}}(\boldsymbol{x}_t, t, 0), G_{\theta_{\text{EMA}}}(\hat{\boldsymbol{x}}_t, t, 0)) \tag{29}$$

which compares the perceptual distance of samples projected to time $t = 0$. In contrast, following iCMs, we use the pseudo-huber loss

$$d(\boldsymbol{x}_t, \hat{\boldsymbol{x}}_t) = \sqrt{\|\boldsymbol{x}_t - \hat{\boldsymbol{x}}_t\|_2^2 + c^2} - c \tag{30}$$

where $c = 0.00054\sqrt{d}$, where $d$ is the dimension of $\boldsymbol{x}_t$.

**Batch size.** We use batch size 128 for $32 \times 32$ resolution images and batch size 64 for $64 \times 64$ resolution images.

**Optimizer.** We use the Adam optimizer (Kingma & Ba, 2015) with learning rate

$$\eta = 0.0002/(128/\texttt{batch\_size}) \tag{31}$$

and default $(\beta_1, \beta_2) = (0.9, 0.999)$.

**Coefficient for $\mathcal{L}_{\text{FM}}(\theta)$.** We use $\lambda_{\text{FM}} = 0.1$ for all experiments.

**Network.** We modify `SongUNet` provided at `https://github.com/NVlabs/edm` to accept two time conditions $t$ and $s$ by using two time embedding layers.

**ODE Solver.** We use the second order Heun solver to calculate $\mathcal{L}_{\text{GCTM}}(\theta)$.

**Gaussian perturbation.** We apply a Gaussian perturbation from a normal distribution multiplied by 0.05 to sample $\boldsymbol{x}_1$, excluding inpainting task.

A.2 EVALUATION

In this section, we describe the details of the evaluation to ensure reproducibility of our experiments.

**Datasets.** In unconditional generation task, we compare our GCTM generation performance using CIFAR10 training dataset. In image-to-image translation task, we evaluate the performance of models using test sets of Edges→Shoes, Night→Day, Facades from Pix2Pix. In image restoration task, we use FFHQ and apply following corruption operators $\boldsymbol{H}$ from I²SB to obtain measurement: bicubic super-resolution with a factor of 2, Gaussian deblurring with $\sigma = 0.8$, and center inpainting with Gaussian. We then assess model performance using test dataset.

**Baselines.** For I2I translation tasks ($64 \times 64$ resolution), we compare three baselines: Pix2Pix from `https://github.com/junyanz/pytorch-CycleGAN-and-pix2pix`, Palette model from `https://github.com/Janspiry/Palette-Image-to-Image-Diffusion-Models`, and I²SB from `https://github.com/NVlabs/I2SB`. In the case of Pix2Pix, during training, we optimize the generator and discriminator with $256 \times 256$ resolution images using the recommended hyper-parameters. During inference, we apply bilinear interpolation to generator output images to resize them to $64 \times 64$. For other baselines, we modify the model image resolution to $64 \times 64$ and use the recommended hyper-parameters. Same configuration is used in supervised image restoration task.

**Metrics details.** We calculate FID using `https://github.com/mseitzer/pytorch-fid` and IS from `https://github.com/pytorch/vision/blob/main/torchvision/models/inception.py`. We assess LPIPS from `https://github.com/richzhang/PerceptualSimilarity` with AlexNet version 0.1. In generation task, we employ the entire training dataset to obtain FID scores, and in the other task, we sample 5,000 test datasets. To obtain PSNR and SSIM, we convert the data type of model output to `uint8` and normalize it. We use `https://github.com/scikit-image/scikit-image` for PSNR and SSIM.

**Sampling time.** To compare inference speed, we measure the average time between the model taking in one batch size as input and outputting it.

# B ALGORITHMS

## B.1 OPTIMAL TRANSPORT

---
**Algorithm 3** Sinkhorn-Knopp (SK)

---
1: **Input:** $\{\boldsymbol{x}_0^m\}_{m=1}^M, \{\boldsymbol{x}_1^m\}_{m=1}^M, \tau$
2: Compute cost matrix $\boldsymbol{C} \in \mathbb{R}^{M \times M}$ subject to $C_{i,j} = \|\boldsymbol{x}_0^i - \boldsymbol{x}_1^j\|_2^2$
3: Solve $\boldsymbol{P}^{\text{EOT}} = \arg\min_{\boldsymbol{P}} \langle \boldsymbol{P}, \boldsymbol{C} \rangle - \tau H(\boldsymbol{P})$ such that $\boldsymbol{P}\boldsymbol{1} = \boldsymbol{P}^\top \boldsymbol{1} = \frac{1}{n}\boldsymbol{1}$ with Algorithm 1 in Cuturi (2013)
4: Treat $\boldsymbol{P}^{\text{EOT}}$ as a discrete distribution over $\{1, \dots, M\} \times \{1, \dots, M\}$
5: Sample $\{(i^m, j^m)\}_{m=1}^M \sim \boldsymbol{P}^{\text{EOT}}$
6: **Return:** $\{(\boldsymbol{x}_0^{i^m}, \boldsymbol{x}_1^{j^m})\}_{m=1}^M$

---

## B.2 IMAGE RESTORATION

In Alg. 4, we describe three zero-shot image restoration algorithms, DPS, CM, and GCTM. DPS uses the posterior mean $\mathbb{E}_{q(\boldsymbol{x}_0|\boldsymbol{x}'_{t_i})}[\boldsymbol{x}_0]$ to both traverse to a smaller time $t_{i-1}$ and to approximate measurement inconsistency. As the posterior mean generally do not lie in the data domain, using it to calculate measurement inconsistency can be problematic. Indeed, approximation error in DPS is closely related to the discrepancy between the posterior mean and $\boldsymbol{x}'_{t_i \to 0}$ (see Theorem 1 in (Chung et al., 2022) for a formal statement). On the other hand, CM uses the ODE terminal point $\boldsymbol{x}'_{t_i \to 0}$ to traverse to a smaller time $t_{i-1}$ and to approximate measurement inconsistency. While CM can have better guidance gradients as $\boldsymbol{x}'_{t_i \to 0}$ lie within the data domain, using $\boldsymbol{x}'_{t_i \to 0}$ to traverse to $t_{i-1}$ can accumulate truncation error and degrade sample quality. For instance, see Figure 9 (a) in (Kim et al.,

---

**Algorithm 4** Zero-shot Image Restoration

---

1: **Input:** Measurement $\boldsymbol{x}_1$, corruption $\boldsymbol{H}$, discretization $\{t_i\}_{i=0}^M$
2: $\boldsymbol{x}'_{t_M} \sim \mathcal{N}(0, \boldsymbol{I})$
3: **for** $i = M$ **to** $1$ **do**
4:     $\boldsymbol{\epsilon} \sim \mathcal{N}(0, \boldsymbol{I})$
5:     **if** Method is DPS **then**
6:         $\hat{\boldsymbol{x}}_0 = g_\theta(\boldsymbol{x}'_{t_i}, t_i, t_i)$
7:         $\boldsymbol{x}'_{t_{i-1}} = (1 - t_{i-1})\hat{\boldsymbol{x}}_0 + t_{i-1}\boldsymbol{\epsilon}$
8:     **else if** Method is CM **then**
9:         $\hat{\boldsymbol{x}}_0 = G_\theta(\boldsymbol{x}'_{t_i}, t_i, 0)$
10:        $\boldsymbol{x}'_{t_{i-1}} = (1 - t_{i-1})\hat{\boldsymbol{x}}_0 + t_{i-1}\boldsymbol{\epsilon}$
11:     **else if** Method is GCTM **then**
12:        Evaluate score and ODE endpoint in parallel by $t = (t_i, t_i)$, $s = (t_i, 0)$ :
13:        $\widetilde{\boldsymbol{x}}_0, \hat{\boldsymbol{x}}_0 = g_\theta(\boldsymbol{x}'_{t_i}, t_i, t_i), G_\theta(\boldsymbol{x}'_{t_i}, t_i, 0)$
14:        $\boldsymbol{x}'_{t_{i-1}} = (1 - t_{i-1})\widetilde{\boldsymbol{x}}_0 + t_{i-1}\boldsymbol{\epsilon}$
15:     **end if**
16:     $\boldsymbol{x}'_{t_{i-1}} \leftarrow \boldsymbol{x}'_{t_{i-1}} - \lambda \nabla_{\boldsymbol{x}'_{t_i}} \|\boldsymbol{x}_1 - \boldsymbol{H}\hat{\boldsymbol{x}}_0\|_2^2$
17: **end for**
18: **Return:** $\boldsymbol{x}'_0$

---

2024b). GCTM mitigates both problems by enabling parallel evaluation of posterior mean and ODE endpoint, as shown in Line 12-13 of Alg. 4.

### B.3 IMAGE EDITING

---

**Algorithm 5** Image Editing

---

1: **Input:** $(\boldsymbol{x}_0, \boldsymbol{x}_1) \sim q(\boldsymbol{x}_0, \boldsymbol{x}_1)$, $t$
2: $\hat{\boldsymbol{x}}_t = (1 - t)\text{Edit}(\boldsymbol{x}_0) + t\boldsymbol{x}_1$
3: **Return:** $G_\theta(\hat{\boldsymbol{x}}_t, t, 0)$

---

## C PROOFS

### C.1 CORRECTNESS OF SECTION 3.3

We show that our exposition in Section 3.3 adheres to Conditional Flow Matching (CFM) theory. Specifically, the notations

$$z, \quad q(z), \quad p_t(x|z), \quad u_t(x|z) \tag{32}$$

in Section 3 of Tong et al. (2024) are expressed in our paper as

$$(\boldsymbol{x}_0, \boldsymbol{x}_1), \quad q(\boldsymbol{x}_0, \boldsymbol{x}_1), \quad q(\boldsymbol{x}_t|\boldsymbol{x}_0, \boldsymbol{x}_1), \quad \boldsymbol{x}_1 - \boldsymbol{x}_0, \tag{33}$$

respectively. It follows that $p_t(x)$ and $u_t(x)$ in Tong et al. (2024) are expressed in our notation as

$$p_t(x) := \int p_t(x|z)q(z)\,dz = \int q(\boldsymbol{x}_t|\boldsymbol{x}_0, \boldsymbol{x}_1)q(\boldsymbol{x}_0, \boldsymbol{x}_1)\,d\boldsymbol{x}_0\,d\boldsymbol{x}_1 = q(\boldsymbol{x}_t) \tag{34}$$

and

$$u_t(x) := \mathbb{E}_{q(z)} \frac{u_t(x|z)p_t(x|z)}{p_t(x)} = \mathbb{E}_{q(\boldsymbol{x}_0, \boldsymbol{x}_1)} \frac{(\boldsymbol{x}_1 - \boldsymbol{x}_0)q(\boldsymbol{x}_t|\boldsymbol{x}_0, \boldsymbol{x}_1)}{q(\boldsymbol{x}_t)} = \mathbb{E}_{q(\boldsymbol{x}_0, \boldsymbol{x}_1|\boldsymbol{x}_t)}[\boldsymbol{x}_1 - \boldsymbol{x}_0] \tag{35}$$

respectively. By Theorem 3.1 in Tong et al. (2024), the ODE Eq. (11) indeed generates $q(\boldsymbol{x}_t)$, and Eq. (12) is equivalent to the CFM objective Eq. (10) in Tong et al. (2024).

## C.2 Proof of Proposition 1

*Proof.* We observe that the velocity term in (11) may be expressed as

$$\mathbb{E}_{q(\boldsymbol{x}_0,\boldsymbol{x}_1|\boldsymbol{x}_t)}[\boldsymbol{x}_1 - \boldsymbol{x}_0] = \mathbb{E}_{q(\boldsymbol{x}_0,\boldsymbol{x}_1|\boldsymbol{x}_t)}[(\boldsymbol{x}_t - \boldsymbol{x}_0)/t] \tag{36}$$

$$= \mathbb{E}_{q(\boldsymbol{x}_0|\boldsymbol{x}_t)}[(\boldsymbol{x}_t - \boldsymbol{x}_0)/t] \tag{37}$$

$$= (\boldsymbol{x}_t - \mathbb{E}_{q(\boldsymbol{x}_0|\boldsymbol{x}_t)}[\boldsymbol{x}_0])/t \tag{38}$$

since $\boldsymbol{x}_1$ is determined given $\boldsymbol{x}_0$ and $\boldsymbol{x}_t$. This shows the equivalence between (11) and (13). Eq. (14) is then a straightforward consequence of the equivalence between ODEs. □

## C.3 Proof of Proposition 2

*Proof.* We first show equivalence of scores. We note that

$$\boldsymbol{x}_\tau \mapsto \boldsymbol{x}_t \tag{39}$$

is a bijective transformation, so by change of variables,

$$q(\boldsymbol{x}_t|\boldsymbol{x}_0) = (1+\tau) \cdot \mathcal{N}(\boldsymbol{x}_\tau|\boldsymbol{x}_0, \tau\boldsymbol{I}) = (1+\tau) \cdot p(\boldsymbol{x}_\tau|\boldsymbol{x}_0) \tag{40}$$

and marginalizing out $\boldsymbol{x}_0$, we get

$$q(\boldsymbol{x}_t) = (1+\tau) \cdot p(\boldsymbol{x}_\tau). \tag{41}$$

It follows by Bayes' rule that

$$p(\boldsymbol{x}_0|\boldsymbol{x}_\tau) = \frac{p(\boldsymbol{x}_\tau|\boldsymbol{x}_0)p(\boldsymbol{x}_0)}{p(\boldsymbol{x}_\tau)} \tag{42}$$

$$= \frac{(1+\tau)^{-1}q(\boldsymbol{x}_t|\boldsymbol{x}_0)q(\boldsymbol{x}_0)}{(1+\tau)^{-1}q(\boldsymbol{x}_t)} \tag{43}$$

$$= \frac{q(\boldsymbol{x}_t|\boldsymbol{x}_0)q(\boldsymbol{x}_0)}{q(\boldsymbol{x}_t)} \tag{44}$$

$$= q(\boldsymbol{x}_0|\boldsymbol{x}_t) \tag{45}$$

and thus

$$\mathbb{E}_{p(\boldsymbol{x}_0|\boldsymbol{x}_\tau)}[\boldsymbol{x}_0] = \mathbb{E}_{q(\boldsymbol{x}_0|\boldsymbol{x}_t)}[\boldsymbol{x}_0]. \tag{46}$$

for all $\tau \in (0, \infty)$ and $\boldsymbol{x}_\tau$. We now show the equivalence of ODEs. Diffusion PFODE is

$$d\boldsymbol{x}_\tau = \frac{\boldsymbol{x}_\tau - \mathbb{E}_{p(\boldsymbol{x}_0|\boldsymbol{x}_\tau)}[\boldsymbol{x}_0]}{\tau} \, d\tau. \tag{47}$$

With the change of variable

$$\boldsymbol{x}_t = \boldsymbol{x}_\tau/(1+\tau), \tag{48}$$

we have

$$d\boldsymbol{x}_t = -\frac{\boldsymbol{x}_\tau}{(1+\tau)^2} \, d\tau + \frac{1}{1+\tau} \, d\boldsymbol{x}_\tau \tag{49}$$

$$= -\frac{\boldsymbol{x}_\tau}{(1+\tau)^2} \, d\tau + \frac{\boldsymbol{x}_\tau - \mathbb{E}_{p(\boldsymbol{x}_0|\boldsymbol{x}_\tau)}[\boldsymbol{x}_0]}{\tau(1+\tau)} \, d\tau \tag{50}$$

$$= -\frac{\boldsymbol{x}_t}{1+t} \, d\tau + \frac{(1+t)\boldsymbol{x}_t - \mathbb{E}_{p(\boldsymbol{x}_0|\boldsymbol{x}_\tau)}[\boldsymbol{x}_0]}{t(1+t)} \, d\tau \tag{51}$$

$$= \frac{\boldsymbol{x}_t - \mathbb{E}_{p(\boldsymbol{x}_0|\boldsymbol{x}_\tau)}[\boldsymbol{x}_0]}{\tau(1+\tau)} \, d\tau \tag{52}$$

$$= \frac{\boldsymbol{x}_t - \mathbb{E}_{q(\boldsymbol{x}_0|\boldsymbol{x}_t)}[\boldsymbol{x}_0]}{\tau(1+\tau)} \, d\tau \tag{53}$$

where we have used equivalence of scores at the last line. We then make the change of time variable

$$t = \tau/(1+\tau) \implies dt = \frac{1}{(1+\tau)^2} \, d\tau \tag{54}$$

which gives us

$$d\boldsymbol{x}_t = \frac{\boldsymbol{x}_t - \mathbb{E}_{q(\boldsymbol{x}_0|\boldsymbol{x}_t)}[\boldsymbol{x}_0]}{\tau/(1+\tau)} \, dt \tag{55}$$

$$= \frac{\boldsymbol{x}_t - \mathbb{E}_{q(\boldsymbol{x}_0|\boldsymbol{x}_t)}[\boldsymbol{x}_0]}{t} \, dt. \tag{56}$$

For the first equality in (20), transform PFODE variables $(\boldsymbol{x}_\tau, \tau)$ into FM ODE variables $(\boldsymbol{x}_t, t)$ with (17), transport $\boldsymbol{x}_t$ to $\boldsymbol{x}_s$ with $G_{\text{GCTM}}$, and then transform FM ODE variables $(\boldsymbol{x}_s, s)$ into PFODE variables $(\boldsymbol{x}_\sigma, \sigma)$ with the inverse of (17). Second equality in (20) follows directly from (18). $\qquad \square$

## D LIMITATION, SOCIAL IMPACTS, AND REPRODUCIBILITY

**Limitations.** GCTMs are yet unable to reach state-of-the-art unconditional generative performance. We speculate further tuning of hyper-parameters in the manner of iCMs could improve the performance, and leave this for future work.

**Social impacts.** GCTM generalizes CTM to achieve fast translation between any two distributions. Hence, GCTM may be used for beneficial purposes, such as fast medical image restoration. However, GCTM may also be used for malicious purposes, such as generation of malicious images, and this must be regulated.

**Reproducibility.** We open-source our code at `https://github.com/1202kbs/GCTM` including training code for unconditional generation, image-to-image translation, and supervised image restoration models.

## E ADDITIONAL EXPERIMENTS

### E.1 COMPARING I2I PERFORMANCE WITH OTHER BASELINE MODELS

We compare the image-to-image (I2I) performance of our model with two baseline approaches: EGSDE (Zhao et al., 2022) and BBDM (Li et al., 2023). Since BBDM, an I2I framework based on the Brownian Bridge process, leverages a latent diffusion model, we train it with a pixel-space diffusion model for a fair comparison. Both BBDM and EGSDE are trained on the Edges→Shoes dataset. As shown in Table 4, our GCTM outperforms all baselines across various metrics, even when evaluated with fewer sampling steps.

In addition, we visualize the image editing results in Fig. 9. While EGSDE generates realistic images, it fails to faithfully preserve the given conditions. BBDM, on the other hand, struggles to perform robustly on (unseen) conditional images. In contrast, GCTM produces realistic images while accurately maintaining the original conditions.

| Method | NFE | Time (ms) | FID ↓ | IS ↑ | LPIPS ↓ |
|---|---|---|---|---|---|
| BBDM (Li et al., 2023) | 5 | 75 | 43.7 | 3.43 | 0.099 |
| EGSDE (Zhao et al., 2022) | 500 | 2590 | 198.1 | 2.87 | 0.476 |
| GCTM | 1 | 87 | **40.3** | **3.54** | **0.097** |

Table 4: Evaluation of I2I translation on Edges→Shoes with other baslines.

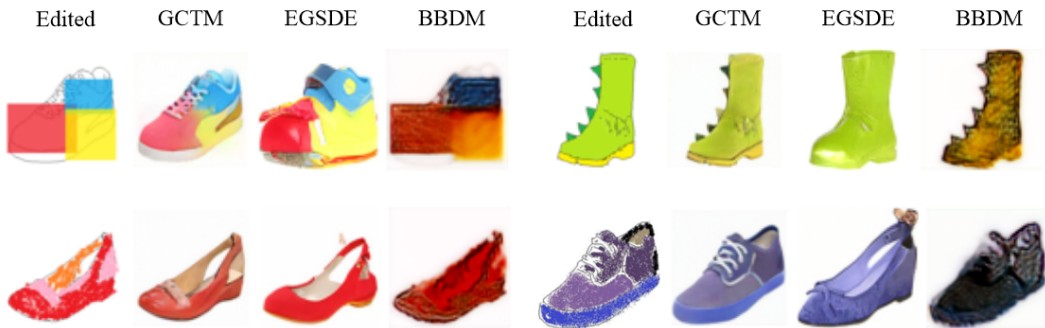

Figure 9: Comparison on image editing with GCTM and other baselines

## E.2 CONTROLLABLE IMAGE EDITING

In this section, we demonstrate that effectiveness of image editing can be controlled. In Algorithm 5, we control the time point $t$ to determine how much of the edited image to reflect. In Fig. 10, the results visualize how $t$ effect the output of model output. We observe that the larger $t$, the more realistic the image, and the smaller $t$, the more faithful the edit feature. We set $t = 0.95$ and $t = 0.4$ at supervised coupling and independent coupling, respectively.

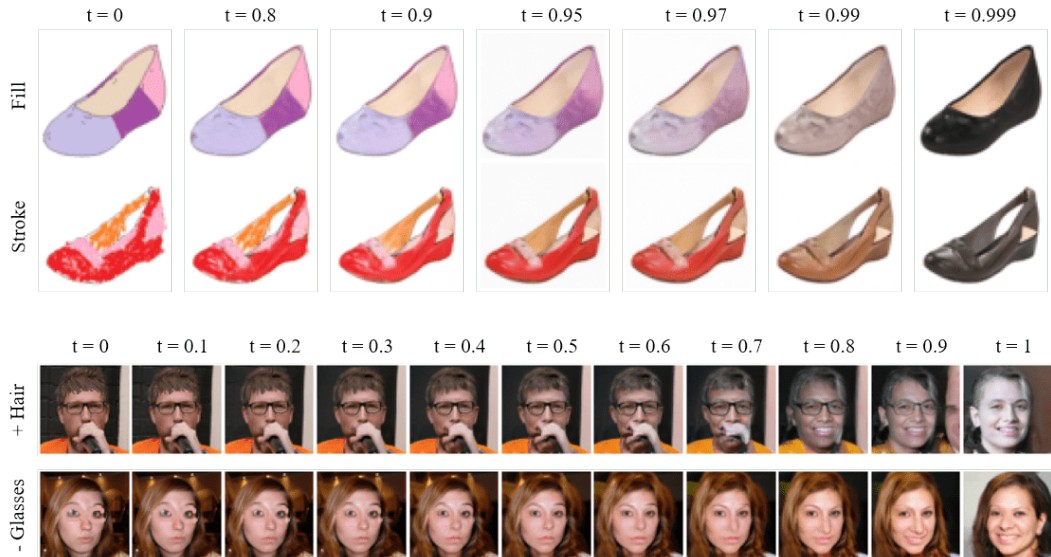

Figure 10: Controllability of image editing by $t$.

### E.3 High-resolution Image-to-Image Translation

To verify the robustness of our framework to scalability of resolution, we experiment with image-to-image translation of the Facades dataset with $256 \times 256$ resolution (Fig. 11). In Table 5, we see that GCTM achieves high performance despite using fewer NFEs compared to the baselines, generating realistic, diverse, and faithful translated images.

| Method | NFE | Facades-256 | | |
| --- | --- | --- | --- | --- |
| | | FID ↓ | IS ↑ | LPIPS ↓ |
| Pix2Pix (Isola et al., 2017) | 1 | 117.2 | 1.60 | **0.414** |
| Palette (Saharia et al., 2022) | 5 | 396.7 | 1.14 | 1.089 |
| I²SB(Liu et al., 2023) | 5 | 128.6 | 2.23 | 0.454 |
| GCTM | 1 | **107.0** | **2.24** | 0.426 |

Table 5: Quantitative evaluation of I2I translation with $256 \times 256$ resolution images. Best is in **bold**, and second best is underlined.

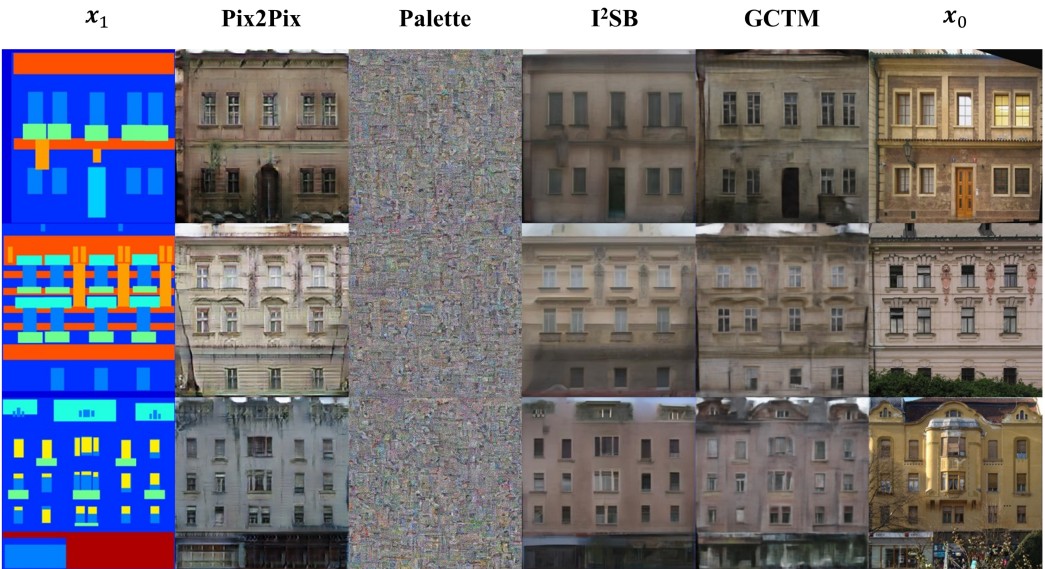

Figure 11: Qualitative comparison of I2I task on Facades $256 \times 256$.

### E.4 HIGH-RESOLUTION IMAGE RESTORATION

In Table 6, we demonstrate image restoration task of GCTM on ImageNet with higher resolution images. As diffusion-based solvers for inverse problems, both DPS (Chung et al., 2022) and DDS (Chung et al., 2024) require a sufficient number of NFEs to achieve effective reconstruction. Specifically for DPS, using significantly fewer NFEs than the 1000 NFEs suggested in the original paper, combined with the absence of measurement noise, disrupts the reconstruction process, resulting in outputs that are worse than GCTM. Although DDS shows improved performance compared to DPS, it requires more NFEs to achieve comparable performance with GCTM, which emphasizes the efficiency of GCTM in solving image restoration tasks.

| Method | NFE | SR4 - Bicubic | | | Deblur - Gaussian | | | Inpaint - Center | | |
|---|---|---|---|---|---|---|---|---|---|---|
| | | PSNR ↑ | SSIM ↑ | LPIPS ↓ | PSNR ↑ | SSIM ↑ | LPIPS ↓ | PSNR ↑ | SSIM ↑ | LPIPS ↓ |
| DPS | 10 | 10.37 | 0.357 | 0.727 | 10.27 | 0.256 | 0.830 | 9.98 | 0.247 | 0.841 |
| | 50 | 16.15 | 0.392 | 0.654 | 19.19 | 0.520 | 0.523 | 13.61 | 0.526 | 0.522 |
| | 1000 | 22.36 | 0.601 | 0.327 | 26.29 | 0.739 | 0.246 | 18.53 | 0.681 | 0.288 |
| DDS | 10 | 19.79 | 0.569 | 0.491 | 21.12 | 0.634 | 0.394 | 13.09 | 0.503 | 0.531 |
| | 50 | 21.25 | 0.571 | 0.409 | 23.33 | 0.704 | 0.245 | 13.57 | 0.485 | 0.511 |
| GCTM | 1 | **26.70** | **0.771** | **0.223** | **34.65** | **0.948** | **0.032** | **21.56** | **0.808** | **0.229** |

Table 6: GCTM evaluation of image restoration on ImageNet with $256 \times 256$ resolution. Best is in **bold**, and second best is underlined.

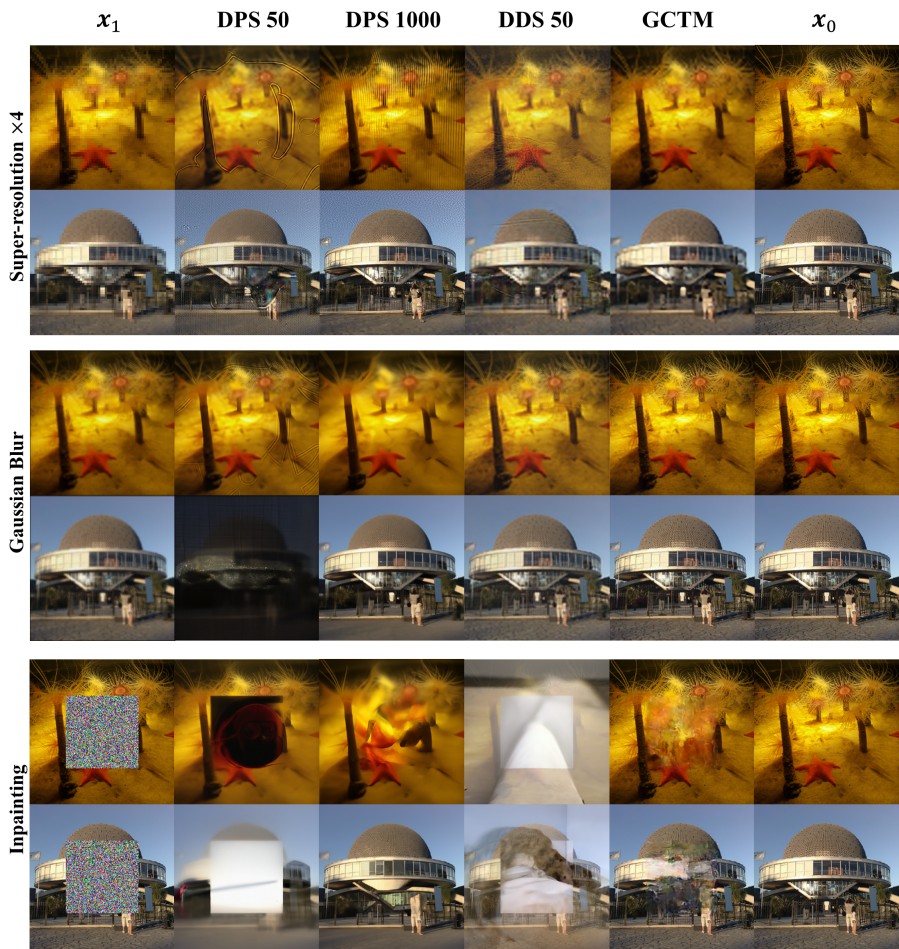

Figure 12: Qualitative comparison of image restoration task on ImageNet 256×256.

### E.5 ADDITIONAL IMAGE-TO-IMAGE TRANSLATION SAMPLES

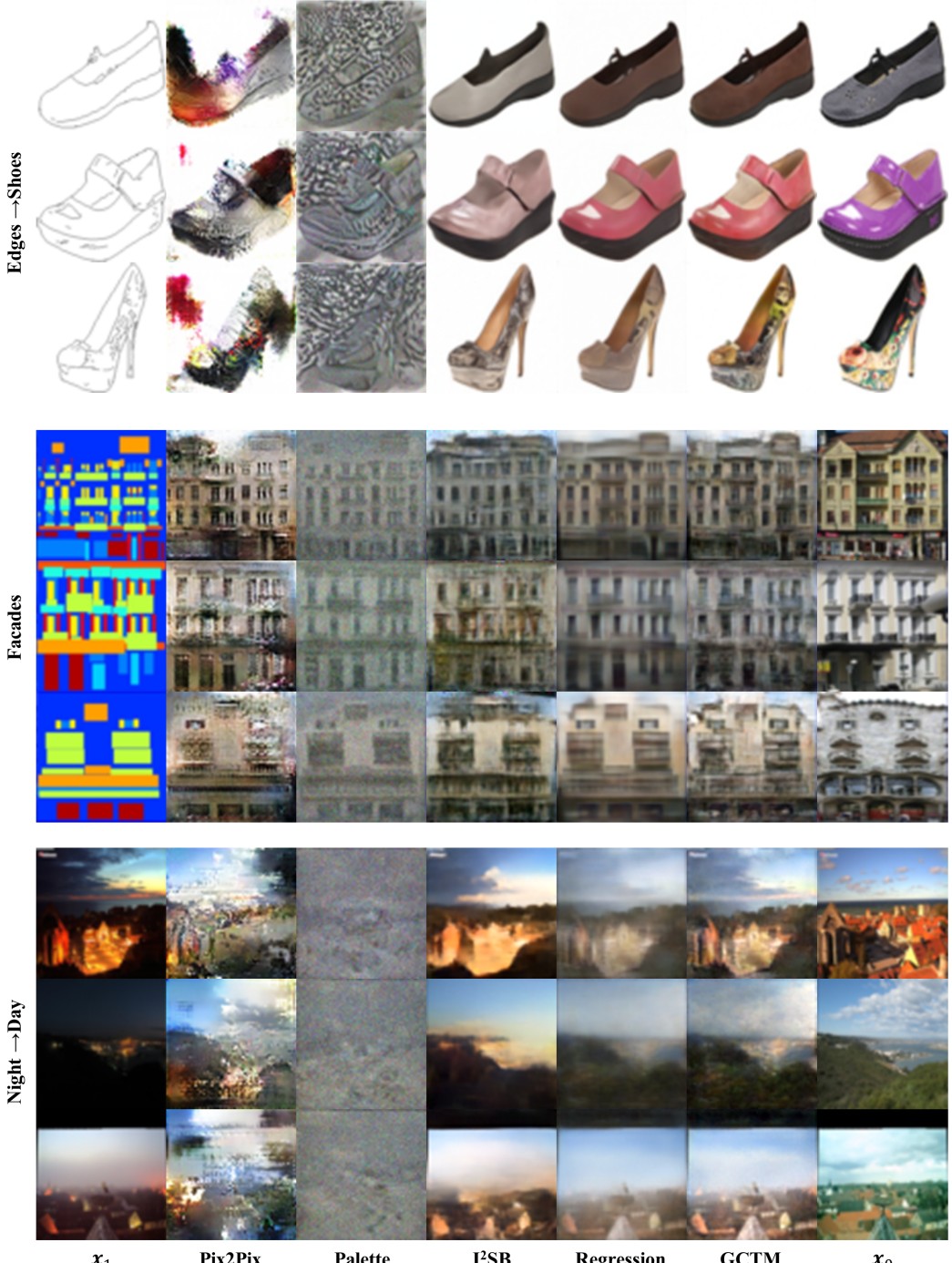

Figure 13: Additional results on image-to-image translation task on Edges→Shoes (top), Facades (middle) and Night→Day (bottom).

### E.6 ADDITIONAL IMAGE RESTORATION SAMPLES

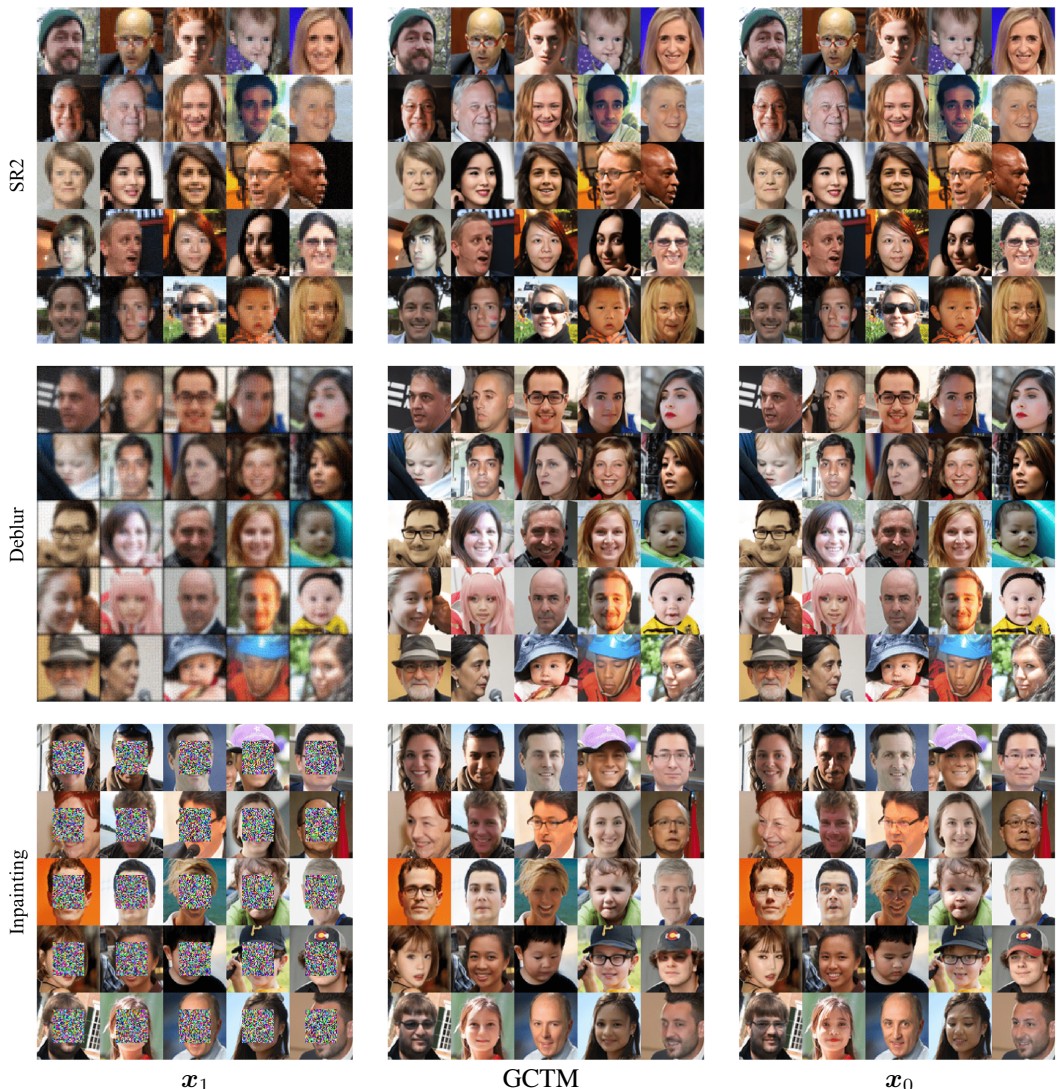

Figure 14: Additional results of supervised image restoration task on FFHQ 64×64.

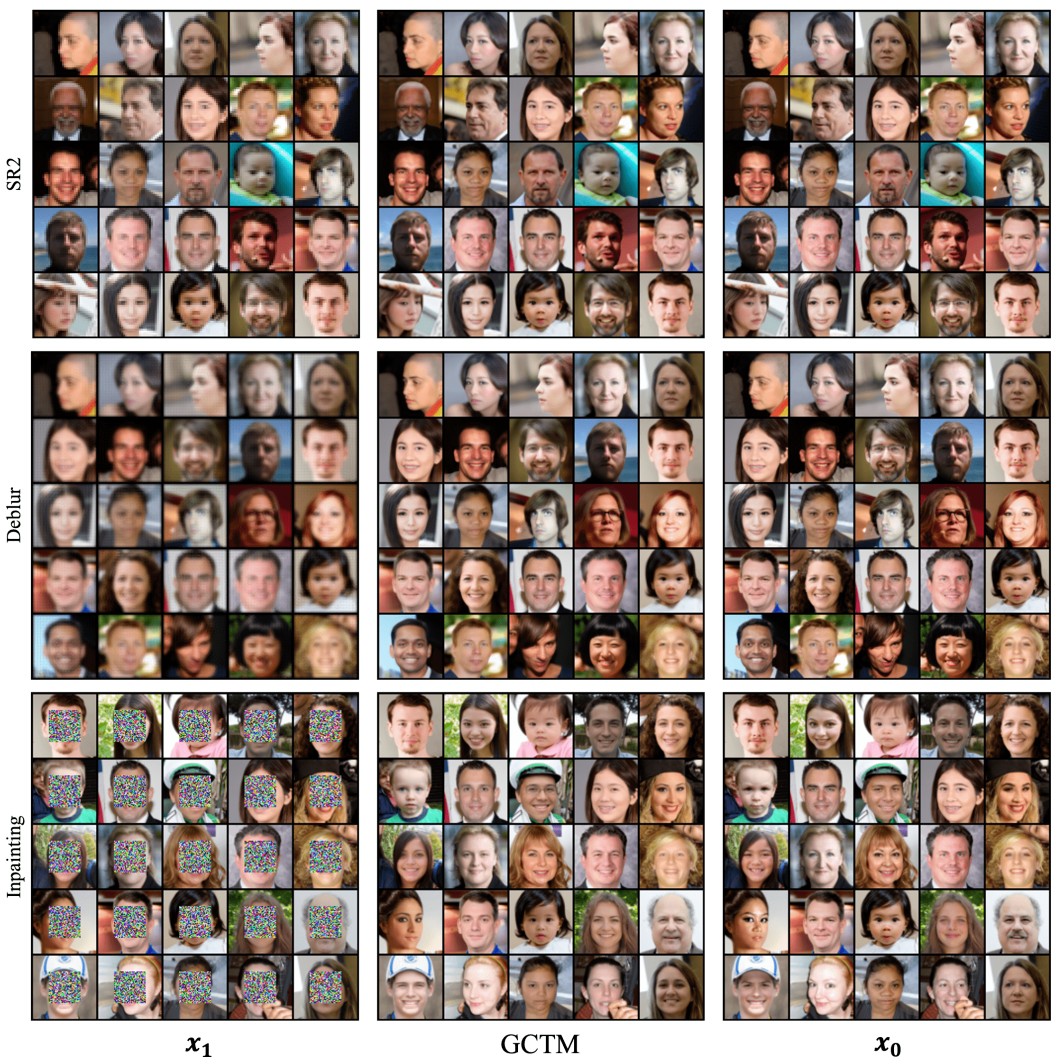

Figure 15: Additional results of zero-shot image restoration task on FFHQ 64×64.

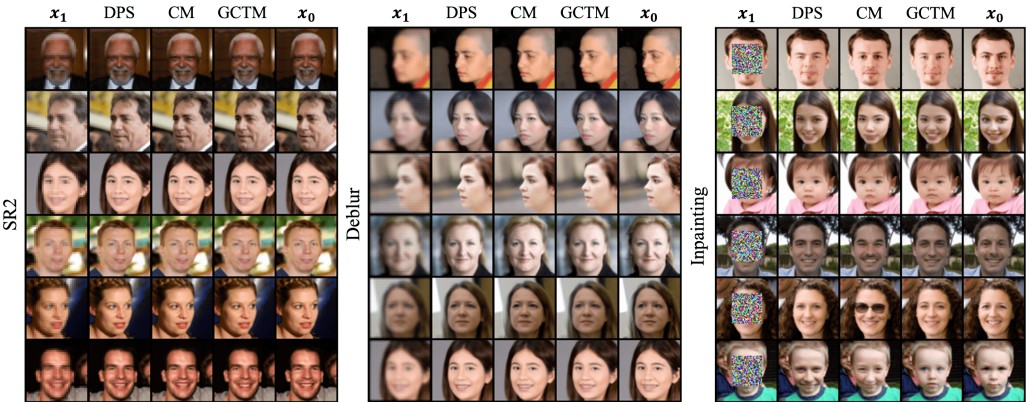

Figure 16: Qualitative comparison of zero-shot algorithms on FFHQ 64×64.

