# OpenReview forum: "Generalized Consistency Trajectory Models for Image Manipulation"
_ICLR.cc/2025/Conference — ICLR 2025 Poster_

### Official Review · Reviewer_PKWr · 2024-11-01

**Soundness:** 2
**Presentation:** 2
**Contribution:** 2
**Rating:** 3
**Confidence:** 4

**Summary:**

This work presents "GENERALIZED CONSISTENCY TRAJECTORY MODELS" a generative model framework that allows consistency trajectory models for arbitrary coupling of image distribution while the original approach of CTM is restricted to the classical (image, pure noise) independent coupling.

**Strengths:**

The paper is a mixed approach of CTM and Conditional Flow Matching that is rigorous.
The theoretical presentation is new as far as I know.

While this is surely of interest, in its current state, the quality of the presentation and the quality of the experiments are not high enough for a pulication at ICLR.

**Weaknesses:**

The main issue of this paper is that CTM is not used on high-resolution images.
The community does not need methods for 64x64 image processing, this resolution is not high enough to assess image to image translation or image restoration.
In comparison,

Unpaired Image-to-Image Translation via Neural Schrödinger Bridge
Download PDF
Beomsu Kim, Gihyun Kwon, Kwanyoung Kim, Jong Chul Ye
ICLR 2024
uses 256x256 and 512x512 and discuss difficulty for other methods to be of higher resolutions.

As another recent example,
Extremal Domain Translation with Neural Optimal Transport
Milena Gazdieva, Alexander Korotin, Daniil Selikhanovych, Evgeny Burnaev
NeurIPS 2023
uses 64x64 and 128x128 examples (already to low in my opinion).

Also comparison with the original pix2pix GAN method is not fair using 64x64 resolution since it is based on a local patch-based discriminator (less resolution means less patches to analyze).


l. 425  "we demonstrate image restoration task of GCTM on ImageNet with higher resolution (256 × 256 resolution) to demonstrate it scalability."
There is no comparison done in the appendix.
Comparison with few NFE and many NFE methods (eg DPS, [Chung & Kim et al ICLR 2023]) is necessary to assess the performance.


All the background section is written without references.
Especially, Section 3.3 FLOW MATCHING (FM) seems far from the original paper by Lipman et al, at least in notation.
Is it just a reinterpretation of CFM using diffusion-like notation?


Another main issue is the use of Optimal Transport coupling (l. 255 (Entropy-regularized) Optimal transport coupling).
This is a chicken and egg problem. If one knew the OT coupling presented by Equation (24) this would be a very good generative model (the best one in a sense) and we would not require a CTM.
What is done in practice is Algorithm 3 Sinkhorn-Knopp (SK) (l. 760) with two batches of size M.
(on a side remark, I don't understand why using entropy regularization here instead of computing an optimal assignment (eg using the emd function from the POT library) unless the batch size M is larger than 1000. This would ensure to use all the data.)
Anyway, there is a big gap between equation (24) which is continuous and discrete algorithm restricted on two discrete batches. Expressing the coupling in continuous space resulting from this discrete procedure is probably intractable (and far from Equation (24)).
This is linked with l. 361: We postulate this is because (1) OT coupling leads to straighter ODE trajectories.


Sampling algorithm and details are missing. This sentence is hard to understand "especially when we use a smaller number of timesteps N (l. 323)". Is gamma sampling from [Kim, Lai et al 2024] used?


Table 1: The CTM paper [Kim, Lai et al 2024] reports FID of 1.98 for Diffusion Models – Distillation Sampling with NFE = 1. What is the 5.28 FID reported here?


The Theorems are more Propositions since their proofs only involves reparameterization of ODE/expectation.


Minor corrections / suggestions:
* l. 170 eq (9): There is a cohabitation of [0,T] and [0,1] convention (change x1 in xT)
* l. 245: joint distributions of q(x0) and q(x1) (speak of q(x0,x1) instead ?)
* l. 253:  add with normal distribution
* l. 269 is hard to inderstands. An ODE as t tends to 0 does not have a rigorous meaning.
* Many references point to ArXiv papers while published at conferences or journals (l. 544, l. 547, l. 553, ...)

**Questions:**

What is the value for the batch size M used for OT-based couplings ?

Table 1: The CTM paper [Kim, Lai et al 2024] reports FID of 1.98 for Diffusion Models – Distillation Sampling with NFE = 1. What is the 5.28 FID reported here?

---

> ### Author Response · Authors · 2024-11-23
> **Reply to Reviewer PKWr (1)**
>
> **[W1] The main issue of this paper is that CTM is not used on high-resolution images. The community does not need methods for 64x64 image processing, this resolution is not high enough to assess image to image translation or image restoration.**
>
> **In comparison, Unpaired Image-to-Image Translation via Neural Schrödinger Bridge Download PDF Beomsu Kim, Gihyun Kwon, Kwanyoung Kim, Jong Chul Ye ICLR 2024 uses 256x256 and 512x512 and discuss difficulty for other methods to be of higher resolutions.**
>
> **As another recent example, Extremal Domain Translation with Neural Optimal Transport Milena Gazdieva, Alexander Korotin, Daniil Selikhanovych, Evgeny Burnaev NeurIPS 2023 uses 64x64 and 128x128 examples (already to low in my opinion).**
>
> **l. 425 "we demonstrate image restoration task of GCTM on ImageNet with higher resolution (256 × 256 resolution) to demonstrate it scalability." There is no comparison done in the appendix. Comparison with few NFE and many NFE methods (eg DPS, [Chung & Kim et al ICLR 2023]) is necessary to assess the performance.**
>
> To address the Reviewer’s concern, we have run image-to-image translation and image restoration experiments on higher resolution (256x256) datasets. Specifically, in Appendix E.3, we provide results on label to facade translation, and show that GCTM consistently does better than Pix2Pix, Palette, and I2SB. In Appendix E.4, we compare GCTM with image restoration algorithms DPS and DDS on ImageNet. We again find that GCTM shows significantly better performance for fast restoration.
>
> **[W2] Also comparison with the original pix2pix GAN method is not fair using 64x64 resolution since it is based on a local patch-based discriminator (less resolution means less patches to analyze).**
>
> In our experiments with 64x64 resolution images, we train pix2pix with 256x256 images and resize the generated images via pooling, so we believe the comparison is fair. We have updated the Appendix to emphasize this point.
>
> **[W3] All the background sections are written without references. Section 3.3 FLOW MATCHING (FM) seems far from the original paper by Lipman et al, at least in notation. Is it just a reinterpretation of CFM using diffusion-like notation?**
>
> Yes, it is a reinterpretation of CFM using diffusion-like notation. We have added appropriate citations to the background section, and provided a full proof of the correctness of Section 3.3 in Appendix C.1.
>
> **[W4] Another main issue is the use of Optimal Transport coupling (l. 255 (Entropy-regularized) Optimal transport coupling).**
>
> **This is a chicken and egg problem. If one knew the OT coupling presented by Equation (24) this would be a very good generative model (the best one in a sense) and we would not require a CTM. What is done in practice is Algorithm 3 Sinkhorn-Knopp (SK) (l. 760) with two batches of size M. (On a side remark, I don't understand why using entropy regularization here instead of computing an optimal assignment (e.g. using the EMD function from the POT library) unless the batch size M is larger than 1000. This would ensure to use all the data.) Anyway, there is a big gap between equation (24) which is a continuous and discrete algorithm restricted to two discrete batches. Expressing the coupling in continuous space resulting from this discrete procedure is probably intractable (and far from Equation (24)). This is linked with l. 361: We postulate this is because (1) OT coupling leads to straighter ODE trajectories.**
>
> We would like to clarify that our goal is not to learn OT or Entropic OT (EOT) maps between distributions, but to use minibatch EOT coupling to accelerate training. We have updated Section 4.1 to emphasize this point. Indeed, the work [1] demonstrates in the context of CFM that using minibatch OT or EOT couplings leads to faster model convergence by reducing gradient variance. We demonstrate a similar benefit for GCTM training in Figure 3 -- EOT coupling accelerates convergence relative to the independent coupling by a factor 2.5.
>
> The reason we use EOT, not OT, is because EOT enables efficient approximation of OT. Specifically, in contrast to the OT problem, EOT problem can be solved several orders of magnitude faster [2]. In Figure 3, we use minibatch size of 256 to compute the EOT coupling, and use 128 pairs to update the network.
>
> [1] Pooladian et al., “Multisample Flow Matching: Straightening Flows with Minibatch Couplings”, ICML, 2023.
>
> [2] Cuturi, “Sinkhorn Distances: Lightspeed Computation of Optimal Transport”, NeurIPS, 2013.

---

> ### Author Response · Authors · 2024-11-23
> **Reply to Reviewer PKWr (2)**
>
> **[W5] Sampling algorithm and details are missing. This sentence is hard to understand "especially when we use a smaller number of timesteps N (l. 323)". Is gamma sampling from [Kim, Lai et al 2024] used?**
>
> We do not describe sampling algorithms for unconditional generation because we always generate one-step samples. To be precise, $N$ is the number of points in the discretization of the time unit interval which is used to sample $s, t, u$, and simulate $x_{t \rightarrow u}$ when computing the GCTM loss.
>
> We admit agree that the sentence pointed out by the Reviewer was rather compact., Thus,and to avoid any confusion, we have provided additional description of the role of time discretization and N in GCTM training (Section 4.1, last paragraph).
>
>
> **[W6] Table 1: The CTM paper [Kim, Lai et al 2024] reports FID of 1.98 for Diffusion Models – Distillation Sampling with NFE = 1. What is the 5.28 FID reported here?**
>
> This is our reproduced result for CTM without GAN loss for a fair comparison. Indeed, when compared with Figure 12 (a) in [1], 5.28 FID is not far off from the reported performance of CTM without GAN loss.
>
> [1] Kim et al., “Consistency Trajectory Models: Learning Probability Flow ODE Trajectory of Diffusion”, ICLR, 2024.
>
> **[W7] The Theorems are more Propositions since their proofs only involve reparameterization of ODE/expectation.**
>
> We agree with the Reviewer and have changed Theorems to Propositions.
>
> **Minor corrections / suggestions:**
>
> **l. 170 eq (9): There is a cohabitation of [0,T] and [0,1] convention (change x1 in xT)**
>
> **l. 245: joint distributions of q(x0) and q(x1) (speak of q(x0,x1) instead ?)**
>
> **l. 253: add with normal distribution**
>
> **l. 269 is hard to understand. An ODE as t tends to 0 does not have a rigorous meaning.**
>
> **Many references point to ArXiv papers while published at conferences or journals (l. 544, l. 547, l. 553, ...)**
>
> Thank you, we have revised the text and references accordingly.

---

> ### Author Response · Authors · 2024-11-24
> **Have we addressed your concerns and questions?**
>
> We thank Reviewer PKWr for the detailed review. We have done our best to address all concerns and questions. In particular, in our revised paper,
>
> - we have provided experiment results on higher resolution images (image-to-image translation and image restoration experiments on 256x256 resolution images, **Appendix E.3 & E.4**) demonstrating the scalability of GCTMs,
> - and have clarified any ambiguities regarding Flow Matching (**Section 3.3 & Appendix C.1**) and Optimal Transport coupling  (**lines 251-259**).
>
> Please let us know if there are any outstanding concerns we can address, or if you have any reservations preventing you from increasing the score.

---

> > ### Comment · Reviewer_PKWr · 2024-11-25
> >
> > Thank you for your reply.
> > I acknowledge that numerous corrections and additions improve the current version of the submission.
> > Yet I still have issues:
> >
> > > [W1] 64x64
> >
> > The revised paper still shows as a main result Table 2 on 64x64 resolution. The resolution limitation is not mentioned in the caption. I see here a danger that this becomes a benchmark for the community while it is not an important problem in my opinion.
> >
> > > [W2] Also comparison with the original pix2pix GAN method is not fair using 64x64 resolution since it is based on a local patch-based discriminator (less resolution means less patches to analyze).
> >
> > > In our experiments with 64x64 resolution images, we train pix2pix with 256x256 images and resize the generated images via pooling, so we believe the comparison is fair. We have updated the Appendix to emphasize this point.
> >
> > What do you mean pooling ? Line 716 is not very clear to me. Is that the updated Appendix (not in blue) ?
> >
> >
> > > [W6] Table 1: The CTM paper [Kim, Lai et al 2024] reports FID of 1.98 for Diffusion Models – Distillation Sampling with NFE = 1. What is the 5.28 FID reported here?
> > > This is our reproduced result for CTM without GAN loss for a fair comparison. Indeed, when compared with Figure 12 (a) in [1], 5.28 FID is not far off from the reported performance of CTM without GAN loss.
> >
> > Is this explained in the submission ? Currently when reading the text line 368, it seems that GCTM is better than CTM, but this is only true when not using a GAN loss.

---

> ### Author Response · Authors · 2024-11-25
>
> Dear Reviewer PKWr,
>
> As the deadline for the discussion period is approaching quickly, we would like to kindly remind the reviewer that we are waiting for your response.
>
> In particular, per your request,  we have done extensive experiments with large size images  to confirm the scalability of our method. So your  timely feedback would be highly appreciated.
>
> Best,
> Authors

---

> ### Author Response · Authors · 2024-11-25
>
> Thank you for the reply. Please let us know whether our clarifications below address your concerns.
>
> **[Q1] The revised paper still shows as a main result Table 2 on 64x64 resolution. The resolution limitation is not mentioned in the caption. I see here a danger that this becomes a benchmark for the community while it is not an important problem in my opinion.**
>
> We have updated the **captions of Table 2, Figure 4, and Table 3** to mention that results are on 64x64 resolution datasets.
>
> **[Q2] What do you mean pooling ? Line 716 is not very clear to me. Is that the updated Appendix (not in blue)?**
>
> By pooling, what we meant was bilinear interpolation. We have added a concrete explanation at **lines 715-717**.
>
> **[Q3] Is this explained in the submission? Currently when reading the text line 368, it seems that GCTM is better than CTM, but this is only true when not using a GAN loss.**
>
> Thank you for pointing out the oversight in the writing. In the revised paper, we have explicitly mentioned that this is our reproduced result without GAN loss at **lines 371-372**.

---

> > ### Author Response · Authors · 2024-11-26
> >
> > Dear Reviewer PKWr,
> >
> > As the deadline for the paper revision approaches, we would like to kindly follow up to check if you have any remaining concerns or comments regarding our submission.
> >
> > In response to your feedback, we have revised the paper and clarified the terminology as requested.
> >
> > Thank you once again for your valuable input. We greatly appreciate your attention to our work, and a timely response would be highly appreciated.
> >
> > Best,
> > Authors

---

> > > ### Author Response · Authors · 2024-11-30
> > >
> > > Dear Reviewer PKWr,
> > >
> > > As the discussion period deadline approaches, we would like to kindly follow up on the ongoing discussion regarding our submission.
> > >
> > > We deeply appreciate the time and effort you have devoted to reviewing our work and for providing constructive feedback. Based on your initial comments, we have revised our paper as follows:
> > >
> > > - **Additional Experiments:** We provided results on higher-resolution images (image-to-image translation and image restoration experiments at 256x256 resolution) to demonstrate the scalability of GCTMs (Appendix E.3 and E.4).
> > > - **Clarifications:** We addressed ambiguities related to Flow Matching (Section 3.3 and Appendix C.1) and Optimal Transport coupling (lines 251–259).
> > >
> > > In your follow-up comment, you acknowledged that the majority of your concerns were addressed, stating, “I acknowledge that numerous corrections and additions improve the current version of the submission,” while requesting minor revisions to the text. We have implemented these requested changes in the manuscript.
> > >
> > > However, we noticed that there has been no further response from you over the past few days. While we fully understand the complexity and time constraints of the review process, we wanted to ensure that any remaining issues could be addressed to your satisfaction within the discussion period. If there are additional points or concerns that require clarification, please feel free to let us know.
> > >
> > > Thank you once again for your thoughtful feedback and for your efforts in reviewing our work.
> > >
> > > Best regards,
> > > Authors

---

> > > > ### Comment · Reviewer_PKWr · 2024-11-30
> > > >
> > > > Dear authors,
> > > >
> > > > I will not raise my score for this submission because I still see problems regarding comparative experiments:
> > > >  - I had to dig that you retrained yourself CTM and report lower performance. At least the performance of the original paper should be reported.
> > > >  - You still compare with 64x64 images which required to downgrade the images of other methods you compare with (downsize images or architecture). I would be more convinced with a 256x256 comparison from the start. From your additional experiments this can be conducted and will make the contribution stronger.
> > > >  - Additional experiments (Table 6 appendix E.4) is also debatable. Why use 10 and 50 NFE for DPS while the original paper uses 1000 NFE (quoting this paper Appendix C.5 SAMPLING SPEED "DPS outperforms all the other methods, whereas in the low NFE regime (≤ 100), DDRM takes over"). As I said in my original review "Comparison with few NFE and many NFE methods (eg DPS, [Chung & Kim et al ICLR 2023]) is necessary to assess the performance." This is OK to report lower performance with your  faster approach, but readers may be interested in the drop of performance (speed VS accuracy trade off).
> > > >
> > > > I think this paper needs a major revision and a fairer way of comparing with previous work. It will then be a strong contribution.

---

> ### Author Response · Authors · 2024-12-01
>
> Dear Reviewer PKWr,
>
> Thank you for your comment. We would like to respectfully express our concern regarding the timing of your request for additional experiments. Specifically, we find it unusual that such a request was made after the revision deadline, despite the opportunity to raise these concerns before November 27, following our rebuttal submission on November 23 and subsequent reminders.
>
> We would also like to kindly remind the Reviewer that, as stated in the review guidelines, the period from November 27 to December 2 is an extended discussion phase. During this time, reviewers “can only ask clarifying questions” and “no requests for new experiments are allowed at this time” and “authors are not required to do major experiments.”
>
> However, we still do our best to address the Reviewer’s concerns below.
>
> > **I had to dig that you retrained yourself CTM and report lower performance. At least the performance of the original paper should be reported.**
>
> In contrast to the Reviewer’s apparent misunderstanding, the FID score for CTM without GAN loss reported in [1] is 5.19 (see Table 3 in [1]), which closely aligns with our reproduced result of 5.28 FID. A difference of 0.09 FID is minimal and, in our view, does not constitute a significant concern as suggested. Notably, [2] reports that FID scores typically exhibit a variation of up to 2% due to random seed effects (see Appendix F.1 in [2]). Our result of 5.28 FID falls well within this expected range.
>
> We would also like to clarify the rationale for not implementing GCTM with GAN loss. First, CTM is already a special case of GCTM. Second, as stated in the second sentence of Section 5.1, the purpose of this section is to examine the effect of independent vs. minibatch EOT coupling for GCTM training. Including GAN loss would introduce confounding factors, which would detract from this objective. To ensure a fair comparison, we conducted experiments for CTM without GAN loss.
>
> Therefore, a comparison involving CTM with GAN loss would be akin to an "apples-to-oranges" comparison, which risks obscuring the primary contributions of our work. Furthermore, given the Reviewer’s late request, made after the revision deadline, we find it unreasonable to perform such additional experiments at this stage. Nevertheless, if this comparison remains a priority, we will include the FID score for CTM with GAN loss in a final version of our paper.
>
> [1] Kim et al., “Consistency Trajectory Models: Learning Probability Flow ODE Trajectory of Diffusion”, ICLR, 2024.
>
> [2] Karras et al., “Elucidating the Design Space of Diffusion-Based Generative Models”, NeurIPS, 2022.
>
> > **You still compare with 64x64 images which are required to downgrade the images of other methods you compare with (downsize images or architecture). I would be more convinced with a 256x256 comparison from the start. From your additional experiments this can be conducted and will make the contribution stronger.**
>
> Contrary to your misunderstanding, we have already implemented and reported the results for the 256x256 experiments in Appendix E.3 and E.4 of our revised paper. Thus, we find it inappropriate for the Reviewer to evaluate our submission based on the initial version of the paper, as this undermines the purpose of the rebuttal and revision process.
>
> > **Additional experiments (Table 6 appendix E.4) is also debatable. Why use 10 and 50 NFE for DPS while the original paper uses 1000 NFE (quoting this paper Appendix C.5 SAMPLING SPEED "DPS outperforms all the other methods, whereas in the low NFE regime (≤ 100), DDRM takes over"). As I said in my original review "Comparison with few NFE and many NFE methods (eg DPS, [Chung & Kim et al ICLR 2023]) is necessary to assess the performance." This is OK to report lower performance with your faster approach, but readers may be interested in the drop of performance (speed VS accuracy trade off).**
>
> We kindly remind the Reviewer that the choice of 10 and 50 NFE for DPS and DDS reflects the significant computational overhead these methods incur at higher NFE levels. For example, restoring a single image requires approximately 120 ms for GCTM (NFE=1), 12,500 ms for DPS (NFE=50), and 6,750 ms for DDS (NFE=50)—a difference of up to hundred orders of magnitude. Given these results and our much faster and competitive solution, it is unclear what the Reviewer expects to gain from additional experiments. Furthermore, the primary focus of this paper is the universality of the proposed algorithm, which supports applications across a wide range of domains, including inverse problems. This level of flexibility is not achievable with DPS and DDS, as they are specifically tailored for inverse problems.

---

> > ### Author Response · Authors · 2024-12-01
> >
> > Dear Reviewer PKWr,
> >
> > We would like to remind you again that, during the discussion extension period, “no requests for new experiments are allowed” for reviewers, and  authors are “not required to perform major experiments.”
> >
> > Nonetheless, to address your late request, we have included image restoration results, including DPS with 1000 NFE, in the tables below. Our results demonstrate that GCTM outperforms DPS with 1000 NFE across all three tasks while being approximately 2000 times faster in terms of wall-clock time.
> >
> >
> > | Method | NFE | Time (ms) | PSNR | SSIM | LPIPS |
> > |--|--|--|--|--|--|
> > | DPS | $10$ | $2500$ | $10.37$ | $0.357$ | $0.727$ |
> > | | $50$     | $12500$ | $16.15$ | $0.392$ | $0.654$ |
> > | | $1000$ | $250000$ | $\underline{22.36}$ | $\underline{0.601}$ | $\underline{0.327}$ |
> > | DDS | $10$ | $1350$ | $19.79$ | $0.569$ | $0.491$ |
> > | | $50$ | $6750$ | $21.25$ | $0.571$ | $0.409$ |
> > | GCTM | $1$ | $110$ | $\mathbf{26.70}$ | $\mathbf{0.771}$ | $\mathbf{0.223}$ |
> >
> > *Table 1. SR4-Bicubic image restoration on ImageNet with 256 × 256 resolution. Best is in bold, and second best is underlined.*
> >
> > | Method | NFE | Time (ms) | PSNR | SSIM | LPIPS |
> > |--|--|--|--|--|--|
> > | DPS | $10$ | $2500$ | $10.27$ | $0.256$ | $0.830$ |
> > | | $50$     | $12500$ | $19.19$ | $0.520$ | $0.523$ |
> > | | $1000$ | $250000$ | $\underline{26.29}$ | $\underline{0.739}$ | $0.246$ |
> > | DDS | $10$ | $1350$ | $21.12$ | $0.634$ | $0.394$ |
> > | | $50$ | $6750$ | $23.33$ | $0.704$ | $\underline{0.245}$ |
> > | GCTM | $1$ | $110$ | $\mathbf{34.65}$ | $\mathbf{0.948}$ | $\mathbf{0.032}$ |
> >
> > *Table 2. Deblur-Gaussian image restoration on ImageNet with 256 × 256 resolution. Best is in bold, and second best is underlined.*
> >
> > | Method | NFE | Time (ms) | PSNR | SSIM | LPIPS |
> > |--|--|--|--|--|--|
> > | DPS | $10$ | $2500$ | $9.98$ | $0.247$ | $0.841$ |
> > | | $50$     | $12500$ | $13.61$ | $0.526$ | $0.522$ |
> > | | $1000$ | $250000$ | $\underline{18.53}$ | $\underline{0.681}$ | $\underline{0.288}$ |
> > | DDS | $10$ | $1350$ | $13.09$ | $0.503$ | $0.531$ |
> > | | $50$ | $6750$ | $13.57$ | $0.485$ | $0.511$ |
> > | GCTM | $1$ | $110$ | $\mathbf{21.56}$ | $\mathbf{0.808}$ | $\mathbf{0.229}$ |
> >
> > *Table 3. Inpaint-Center image restoration on ImageNet with 256 × 256 resolution. Best is in bold, and second best is underlined.*

---

### Official Review · Reviewer_AdFa · 2024-11-03

**Soundness:** 3
**Presentation:** 3
**Contribution:** 3
**Rating:** 8
**Confidence:** 2

**Summary:**

This paper proposes Generalized Consistency Trajectory Models (GCTMs), which extend Consistency Trajectory Models (CTMs) to support sampling from arbitrary distributions rather than being restricted to Gaussian distributions. By reparameterizing the flow matching (FM) ODE in a way analogous to CTMs, the authors provide a general framework that includes CTMs as a special case when the sampled distributions is Gaussian. The paper also explores the design space of the proposed method. The proposed method achieves state-of-the-art performance in multiple image generation and manipulation tasks without relying on pre-trained teacher models.

**Strengths:**

* The paper is theoretically sound and inspring in that:
(1) The reparameterization of the FM ODE to resemble the CTM form allows the proposed GCTM to perform consistent trajectory sampling between arbitrary distributions, which is highly flexible and useful.
(2) GCTM is rigorously shown to generalize CTMs. This makes it possible to train the GCTM using the same training stratagies for training CTMs.

* The paper includes exploration of the design space for the proposed model, such as OT coupling for accelerated sampling (~2.5x as the baseline), Gaussian perturbation for more diverse generation.

* GCTM achieves state-of-the-art results across various applications, such as image restoration and translation, without needing a pre-trained teacher model.

**Weaknesses:**

*Gaussian Perturbation in Section 4.1:

The introduction of Gaussian perturbation seems contradictory to the main goal of GCTM - sampling from arbitrary distributions. Specifically, in the case of multiple labels corresponding to a single observation (L302-303), Gaussian perturbation does not apply necessarily, as these label variations are not expected to follow an iid Gaussian distribution.

*Lack of teacher model distillation training scheme:

The reason for the absence of distillation from a teacher network is not fully addressed, although the method is claimed to be a generalization of CTM method. Given that distilling from a teacher could potentially improve training efficiency, an explanation for this choice would strengthen the paper.

*Performance gap with iCM:

GCTM does not outperform improved Consistency Models (iCM), but the reason for this gap is not fully explored. A deeper analysis of this performance difference would be valuable.

**Questions:**

Please refer to the weakness section above.

---

> ### Author Response · Authors · 2024-11-23
> **Reply to Reviewer AdFa**
>
> **[W1] Gaussian Perturbation in Section 4.1:**
>
> **The introduction of Gaussian perturbation seems contradictory to the main goal of GCTM - sampling from arbitrary distributions. Specifically, in the case of multiple labels corresponding to a single observation (L302-303), Gaussian perturbation does not apply necessarily, as these label variations are not expected to follow an iid Gaussian distribution.**
>
> Thanks for the comments that can highlight the advantage of our model. Conditional Flow Matching theory guarantees that GCTM translates the source distribution into the correct target distribution. For instance, consider the extreme case when the corruption operator H is the zero matrix, and we add Gaussian noise of variance 1. Then,
>
> $q(\boldsymbol{x}_0,\boldsymbol{x}_1) = q(\boldsymbol{x}_0) \mathcal{N}(\boldsymbol{x}_1|\boldsymbol{0}, \boldsymbol{I})$
>
> and by Proposition 2 in our paper, GCTM learns to map Gaussian noise to q(x0), despite q(x0) not following an i.i.d. Gaussian distribution.
>
> **[W2] Lack of teacher model distillation training scheme:**
>
> **The reason for the absence of distillation from a teacher network is not fully addressed, although the method is claimed to be a generalization of CTM method. Given that distilling from a teacher could potentially improve training efficiency, an explanation for this choice would strengthen the paper.**
>
> The reason we do not use teachers is simply because most public flow models are trained to generate data from Gaussian noise, but we consider the more general task of translating between arbitrary distributions (e.g., Section 5.2). This naturally motivated us to develop an algorithm which does not require a teacher model.
>
> **[W3] Performance gap with iCM:**
>
> **GCTM does not outperform Improved Consistency Models (iCM), but the reason for this gap is not fully explored. A deeper analysis of this performance difference would be valuable.**
>
> Unlike iCMs, we did not perform an extensive search of GCTM hyper-parameters. Indeed, in Table 1, we see that our GCTM trained without teacher beats vanilla consistency training (CM without teacher), so we believe that with further parameter tuning similar to iCM, we may obtain results that are on par with iCMs. In fact, the prior work [1] demonstrates that by adding a GAN loss to CTM (which is a special case of GCTM by Proposition 2) training, one can achieve state-of-the-art unconditional generative performance (e.g., on CIFAR10, 1.87 FID, which is better than 2.51 FID of iCM).
>
> [1] Kim et al., “Consistency Trajectory Models: Learning Probability Flow ODE Trajectory of Diffusion”, ICLR, 2024.

---

> > ### Comment · Reviewer_AdFa · 2024-11-29
> >
> > My questions have been addressed. I'm willing to keep my rating.

---

> > > ### Author Response · Authors · 2024-11-29
> > > **Thanks!**
> > >
> > > Dear Reviewer AdFa,
> > >
> > > Thanks for your positive comments and maintaining the good rating.
> > > We are glad to hear that your questions have been fully addressed.
> > >
> > > Best,
> > > Authors

---

### Official Review · Reviewer_oui9 · 2024-11-04

**Soundness:** 4
**Presentation:** 3
**Contribution:** 4
**Rating:** 8
**Confidence:** 4

**Summary:**

Consistency trajectory models (CTM) are a recent technique for accelerated sampling of diffusion models, which involves training a model to predict any intermediate point of the Probability Flow ODE (PFODE) trajectory allowing traversal between any time points along the PFODE. This work extends CTMs to develop generalized CTMs (GCTMs) using conditional flow matching to enable one-step translation between two arbitrary distributions via ODEs instead of only Gaussian noise-to-data transformation in CTMs. They utilize flow matching to allow more flexible couplings between starting and target distributions, including independent coupling used by diffusion models as a special case. This broadens the applicability of GCTMs to handle arbitrary image-to-image translation tasks in addition to unconditional generation, and reduces the computational costs associated with these tasks by reducing the need for multiple neural function evaluations.

**Strengths:**

This is overall a nice submission with significant novel technical contributions.

The proposed  generalized CTM is formulated in a principled manner and is explained well.

The paper nicely extends consistency trajectory models to allow translation between arbitrary distributions via flow matching.

The design space is systematically examined, studying the effects of different couplings,  Gaussian perturbation and $\sigma_{max}$ for  stable training.

Experiments are performed on a variety of image restoration and editing tasks in both zero-shot and  supervised manner. In image restoration, the proposed method outperforms consistency models in zero-shot setting, and provides results competitive with supervised regression with a higher perceptual quality.

**Weaknesses:**

I do not find any major concerns in the paper.

In Fig.6, the noise to image GCTM  editing results with NFE = 1 look a little blurred, with background details washed out.

The authors could consider citing and discussing the following parallel works which also aim to perform image-to-image tasks with few NFEs:
Mei etal. CoDi: Conditional Diffusion Distillation for Higher-Fidelity and Faster Image Generation. In CVPR 2024
Zhao etal. CoSIGN: Few-Step Guidance of ConSIstency Model to Solve General INverse Problems. In ECCV 2024
He etal. Consistency Diffusion Bridge Models. In NeurIPS 2024.
Xiao etal. CCM: Real-Time Controllable Visual Content Creation Using Text-to-Image Consistency Models. In ICML 2024
Starodubcev etal. Invertible Consistency Distillation for Text-Guided Image Editing in Around 7 Steps. Arxiv June 2024.

**Questions:**

Can you provide the details of computational costs, and training time required for GCTM, and compare this with  CTM, and the  teacher model?


In CTMs [Kim et al., 2024b], adversarial training can be incorporated to further enhance the quality of samples. Can adversarial training also be incorporated in the proposed GCTMs?

---

> ### Author Response · Authors · 2024-11-23
> **Reply to Reviewer oui9**
>
> **[W1] In Fig.6, the noise to image GCTM editing results with NFE = 1 look a little blurred, with background details washed out.**
>
> We note that in general, there is a trade-off between editing strength and background preservation. For instance, Figure 10 in our paper shows more details in the original image are lost as we increase the amount of added noise. To preserve background detail, we need additional techniques such as attention map manipulation [1,2], but this is out of scope of our work.
>
> [1] Tumanyan et al., “Plug-and-Play Diffusion Features for Text-Driven Image-to-Image Translation”, CVPR, 2023.
>
> [2] Parmar et al., “Zero-shot Image-to-Image Translation”, SIGGRAPH, 2023.
>
> **[W2] The authors could consider citing and discussing the following parallel works which also aim to perform image-to-image tasks with few NFEs:**
>
> **[1] Mei etal. CoDi: Conditional Diffusion Distillation for Higher-Fidelity and Faster Image Generation. In CVPR 2024**
>
> **[2] Zhao etal. CoSIGN: Few-Step Guidance of ConSIstency Model to Solve General INverse Problems. In ECCV 2024**
>
> **[3] He et al. Consistency Diffusion Bridge Models. In NeurIPS 2024**
>
> **[4] Xiao etal. CCM: Real-Time Controllable Visual Content Creation Using Text-to-Image Consistency Models. In ICML 2024**
>
> **[5] Starodubcev etal. Invertible Consistency Distillation for Text-Guided Image Editing in Around 7 Steps. Arxiv June 2024**
>
> We thank the Reviewer for pointing out missing references. We have added them to Section 2 of our revised paper. We also provide a lengthier comparison of our work and the mentioned works below.
>
> Discussion of [1] : this method adds a conditional adaptor on top of a pre-trained latent diffusion model, and optimizes the adaptor to minimize a conditional distillation loss. Unlike our work which distills image-to-image ODE trajectories, [1] distills conditional noise-to-image ODE trajectories.
>
> Discussion of [2] : this method proposes two levels of measurement constraint for solving inverse problems via consistency models. First, the authors add a conditional adaptor on top of a pre-trained consistency model, and optimize the adaptor to predict reconstruction given corrupted measurement as a condition. Second, during generation, the authors provide gradient guidance to minimize measurement error to intermediate samples. In fact, we find that this inference scheme is identical to the zero-shot reconstruction algorithm for CM described in Algorithm 4 of our paper.
>
> Discussion of [3] : this method proposes to learn consistency models along denoising diffusion bridge model (DDBM) ODEs. While this model also achieves a few-step translation from one distribution to another, GCTMs are more general in the sense that they enable velocity evaluation and translation between two arbitrary timesteps.
>
> Discussion of [4] : this method adds a conditional adaptor on top of a pre-trained text-to-image consistency model to enable two levels of control, via image condition and text. Unlike our work which distills image-to-image ODE trajectories, [3] also distills conditional noise-to-image ODE trajectories.
>
> Discussion of [5] : this method trains two consistency distillation models -- a forward model which translates image to noise, and a backward model which translates noise to image. [5] achieves few-step translation by concatenating the forward and backward consistency distillation models.
>
> **[W3] Can you provide the details of computational costs, and training time required for GCTM, and compare this with CTM, and the teacher model?**
>
> Per-iteration costs for CTM and GCTM are identical. We do not use a teacher model, as noted at the beginning of Section 5 our paper. The reason we do not use teachers is simply because most public flow models are trained to generate data from Gaussian noise, but we consider the more general task of translating between arbitrary distributions (e.g., Section 5.2). This naturally motivated us to develop an algorithm which does not require a teacher model.
>
> **[W3] In CTMs [Kim et al., 2024b r], adversarial training can be incorporated to further enhance the quality of samples. Can adversarial training also be incorporated in the proposed GCTMs?**
>
> Yes, adversarial training can also be incorporated into GCTM training in a straightforward manner by maximizing the following loss with respect to the discriminator $d_\eta$ and minimizing with respect to GCTM $G_\theta$
>
> $\mathcal{L_{GAN}}(\theta, \eta) = \mathbb{E_{q(x_0)}} [\log d_\eta(x_0)] + \mathbb{E_{0 \leq s \leq t \leq 1}} \mathbb{E_{q(x_t)}}[\log (1 - d_\eta(G_{sg(\theta)}(G_\theta(x_t,t,s),s,0) ))]$
>
> but we did not explore this option in our work.

---

### Meta-Review · Area_Chair_KJT5 · 2024-12-23

**Metareview:**

This paper addresses the computational inefficiency of the multi-step denoising process in diffusion-based generative models. Its key contribution is the introduction of the Generalized Consistency Trajectory Model (GCTM), designed to flexibly translate between two arbitrary distributions. All reviewers agreed that the paper is theoretically well-grounded and that GCTM represents a highly novel approach. Following the author-review discussion phase, the authors further enhanced the manuscript, particularly by extending the experimental results. The AC noted the negative score provided by Reviewer PKWr, as well as private communications from the authors. Upon careful examination of Reviewer PKWr’s comments, the AC found that the reviewer expressed a positive stance on the theoretical aspects of the paper. Reviewer PKWr also provided valuable feedback, particularly the need for experiments on higher-resolution images. In response, the authors conducted experiments with high-resolution patches in early December. The AC believes this partially satisfies Reviewer PKWr’s recommendation for a major revision and a fairer comparison with prior work. Reviewer PKWr stated, “...a major revision and a fairer way of comparing with previous work. It will then be a strong contribution.”

Given the above considerations, the AC recommends accepting this paper.

**Additional Comments On Reviewer Discussion:**

Please refer to the meta review above.

---

### Decision · Program_Chairs · 2025-01-22

Accept (Poster)